# Validation of the Keen Eye computer-based method for diagnosing visual neglect using a dual-task paradigm

Elizaveta Vasyura[1]*, Maria Kovyazina[1,2,3], Georgiy Stepanov[1,2], Olga Russkikh[4,5], Daria Terentiy[1,2], Victoria Propustina[1,2], Anatoliy Skvortsov[1,2], Nataliya Varako[1,2,3], Svetlana Vasilyeva[2,6], Vadim Daminov[6], Yuri Zinchenko[1,2]

**1** Faculty of Psychology, Lomonosov Moscow State University, Moscow, Russia, **2** Federal Scientific Center of Psychological and Multidisciplinary Research, Moscow, Russia, **3** Research Center of Neurology, Moscow, Russia, **4** State Autonomous Healthcare Institution of the Perm Region "City Clinical Hospital No 4", Perm, Russia, **5** Academician Ye. A. Vagner Perm State Medical University of the Ministry of Healthcare of the Russian Federation, Perm, Russia, **6** Medical Rehabilitation Clinic, Pirogov National Medical and Surgical Center, Moscow, Russia

* vasyuraev@my.msu.ru

## Abstract

Visual neglect is a common and disabling consequence of right hemisphere damage. Standard paper-and-pencil assessments may fail to detect subtle or well-compensated cases of neglect, especially under low attentional demands. This study presents and validates the Keen Eye computer-based method for diagnosing neglect using a dual-task paradigm. The method involves the simultaneous detection of lateralized visual targets and identification of centrally presented digits, increasing attentional load and sensitivity to spatial biases. We tested 102 patients with right hemisphere damage (38 with neglect, 64 without) using a predefined set of target positions that systematically varied across the visual field. Classification models based on error patterns and asymmetry coefficients demonstrated high sensitivity and specificity in distinguishing patients with neglect. The method also revealed quadrant-specific and vertically biased omissions. The findings support the utility of attention-demanding computer-based tasks for improving diagnostic precision in visual neglect and suggest potential for identifying subclinical or hidden neglect profiles.

## Introduction

### Neglect syndrome

Unilateral spatial neglect is a heterogeneous and multi-component syndrome [1] characterized by the lack of conscious perception of stimuli in the space contralateral to the brain lesion, as well as by non-lateralized attention deficits [2]. Neglect syndrome is a common consequence of right hemisphere damage, with prevalence rates

**Data availability statement:** All relevant data are within the paper and its Supporting Information files.

**Funding:** This study was supported by the Ministry of Science and Higher Education of the Russian Federation under the research project 075-15-2024-526. Funder Website: https://minobrnauki.gov.ru Maria S. Kovyazina, Georgy K. Stepanov, Daria D. Terentiy, Victoria A. Propustina, Anatoly A. Skvortsov, Nataliya A. Varako, Svetlana A. Vasilyeva, Yuri P. Zinchenko were supported by this funding. The funder had no role in study design, data collection and analysis, decision to publish, or preparation of the manuscript.

**Competing interests:** The authors have declared that no competing interests exist.

ranging from 43% to 82% among patients in the acute phase of stroke [3–6]. Neither gender nor handedness significantly affects the likelihood of developing neglect following right hemisphere damage [4]. Neglect syndrome manifests across various sensory modalities [1], with visual neglect (VN) being the most functionally limiting for patients. VN significantly impairs daily functioning, affecting essential activities such as eating, personal hygiene, and mobility [7].

Left-sided neglect is a significant negative prognostic factor for post-brain injury functioning. It is reliably associated with poor functional recovery outcomes, and research has shown that the severity of VN serves as an independent predictor of lower recovery outcomes in daily activities after stroke [8,9]. In everyday life, VN has a substantial negative impact on independence in daily activities—patients tend to ignore the left side while reading, writing, navigating, and working [10].

VN following right hemisphere damage is more common than neglect in other sensory modalities [11]. While neglect can manifest in visual, tactile, and auditory modalities—either in isolated or combined forms—its severity is often greater in the visual domain [11,12]. This disproportionate severity may be related to the more significant role of automatic attentional processes toward ipsilateral stimuli in the visual modality compared to tactile and auditory modalities.

VN can manifest in both the horizontal and vertical planes, affecting patients' perception of near and far space [13]. The primary symptoms of VN have been predominantly described in the horizontal plane as an inability to attend to stimuli on the left side of space [14]. When performing visuospatial tests, patients may fail to notice targets on the left side, deviate to the right when bisecting lines, and omit the left side of drawings [15].

Vertical neglect often coexists with horizontal neglect [16,17]. This condition manifests as inattention to stimuli in the upper or lower parts of the visual field. When combined with horizontal neglect, it frequently results in neglect of the lower left portion of the visual field [16,18]. Vertical neglect may also be associated with radial neglect, affecting perception in near or far peripersonal space. Studies have shown that patients with VN are more likely to ignore objects in the lower far space [19]. This phenomenon can be explained by differences in spatial reference frames, with errors in radial and vertical bisection likely stemming from impairments in the retinotopic reference system [20]. Vertical neglect can significantly impact daily activities, such as wheelchair navigation and stair climbing, underscoring the importance of its clinical assessment and treatment [21].

In contemporary neglect classification, a subtype of neglect known as vertical neglect is described [22]. Studies have shown that patients with VN tend to ignore the lower portion of the visual field, as evidenced by multiple studies utilizing visual search and line bisection tasks. For example, one study found that 13 patients with VN following a right hemisphere stroke detected fewer targets and made more fixations in the lower-left quadrant [23]. Similar results were obtained from line cancellation tasks in 23 patients with right-sided stroke and experiments using central cues, where more pronounced deficits were observed in the lower part of the visual field [24]. Deviations upwards were often observed in vertical line bisection tests,

indicating a lack of attention to lower stimuli [25]. In one case, a 72-year-old patient who had experienced a stroke showed an upward deviation, while in another case, quadrant-specific retinotopic defects persisted irrespective of the direction of gaze [26]. These defects are thought to be linked to disruptions in attention networks, with dysfunction in the dorsal attention network associated with vertical neglect, and damage to the right temporal lobe linked to neglect in the upward direction [25].

## Diagnosis of visual neglect

Currently, more than 90 different tests are available for identifying neglect [27]. However, systematic studies indicate that most of these methods lack sufficient validity and reliability [28]. The primary diagnostic approaches include question-naires, behavioral tests, and traditional paper-and-pencil methods.

Traditional paper-and-pencil tests include cancellation tasks, line bisection tasks, and drawing tests. Among these, the most widely used are the Bell's Test [29] and letter cancellation tasks, which have demonstrated high accuracy in detect-ing signs of spatial inattention. However, the line bisection test has limited sensitivity and may fail to identify up to 40% of VN cases [30].

Despite their widespread use, paper-and-pencil methods have low ecological validity, limiting their predictive power for real-world functioning. Additionally, milder forms of VN often go undetected in these tests, and diagnosis may only be possible through the analysis of indirect indicators, such as a right-to-left visual scanning strategy [31].

To address these limitations, more ecologically valid assessment tools have been developed. Among them, the Cather-ine Bergego Scale (CBS) stands out for its reliability, validity, and sensitivity to changes during rehabilitation [32]. Fur-thermore, functional tests, computer-based programs, and virtual reality technologies have proven to be more effective in detecting VN compared to paper-and-pencil tests [33,34]. In addition to their flexibility in task design, computer-based assessments offer several methodological advantages that enhance the quantification of patient performance. First, they enable brief stimulus presentation and millisecond-precision recording of response latencies, allowing the detection of not only overt omissions in contralesional space but also subtle delays in spatial processing. Second, computerized tasks can be adapted to each patient's performance level by adjusting parameters such as stimulus duration, contrast, or spatial location until a threshold is reached, which helps minimize ceiling and learning effects. Finally, computerized environments allow for precise control of attentional load by combining primary and secondary tasks under rigorously timed conditions, thereby increasing diagnostic sensitivity and ensuring reproducibility. While similar attentional demands could theoretically be created without computers, computerized implementation provides superior standardization and accuracy.

Given the advantages and limitations of different diagnostic approaches, a combined diagnostic strategy is recom-mended, incorporating various types of neglect assessments. A comprehensive diagnostic approach integrates traditional paper-and-pencil tests, computer-based tools, and behavioral assessments. This allows for a more precise identifica-tion of VN subtypes while ensuring high ecological validity for both clinical practice and research [35,36]. Thus, modern approaches to VN diagnosis aim for standardization and improved validity, which is particularly important for enhancing diagnostic accuracy and developing effective rehabilitation strategies.

## Dual-task paradigm

Dual-task methods in cognitive psychology trace their origins to early curiosities about the mind's limitations. Pashler and Johnston [37] note that late-nineteenth-century accounts of automatic writing sparked interest in simultaneous task performance. Jersild's 1927 work [38] on task switching provided an early systematic framework, that laid the foundation for later inquiries into multitasking. Resource theories formalized in the mid-twentieth century [39,40], established mod-els of attentional allocation and central processing limits [41]. Nowadays the dual-task paradigm, which encompasses a broad class of methods requiring the simultaneous performance of two tasks—a combination of cognitive/motor, motor/motor, or cognitive/cognitive tasks [42]. The dual-task paradigm is applicable to the study of attention, executive functions

[42] working memory [43], and problem-solving [44]. Performing two tasks simultaneously typically leads to interference effects, reflecting the increased demands on the information-processing system [45]. This paradigm has shown promise in detecting mild cognitive impairments and as a rehabilitation tool [42].

Performing two or more tasks simultaneously, especially when each of the parallel tasks is not highly automated, can lead to cognitive overload. A phenomenon known as "attentional blindness" has been described in the context of visual search, occurring when individuals fail to notice unexpected stimuli while engaged in complex tasks [46]. Various neural indicators of different aspects of attentional blindness have been identified: parieto-occipital potentials reflect deficits in target detection, while frontoparietal coherence signals deficits in discrimination [47]. Numerous studies have reported decreased activity in the temporoparietal regions under working memory overload [48–50]. Attentional blindness also modulates activity in the primary visual cortex, leading to reduced responses in visual areas V3 and, possibly, V2 and V1, accompanied by decreased activation in the inferior parietal cortex [51]. Additionally, according to Lavie's load theory of selective attention [52,53], attentional load can be divided into perceptual and cognitive components, each exerting distinct effects on distractor processing and attentional control. Perceptual load refers to the amount of relevant sensory information to be processed. When perceptual load is high, attentional capacity is exhausted by task-relevant stimuli, which in turn suppresses the processing of irrelevant distractors via passive filtering mechanisms. Conversely, cognitive load—especially load on working memory or executive coordination—impairs top-down attentional control. Under high cognitive load, the active maintenance of attentional priorities is weakened, leading to greater susceptibility to interference from irrelevant stimuli. This dissociation suggests that cognitive and perceptual loads operate via different mechanisms to modulate attentional selection.

Another related phenomenon is **"attentional blink"**—a temporary reduction in attention following the detection of a visual target [54]. This occurs when processing a large amount of rapidly incoming information. Attentional blink is observed when two stimuli are presented in quick succession, typically within 200–500 ms [55,56]. Experiments have demonstrated this phenomenon: if stimuli are presented at high speed in the foveal or peripheral visual field, perception of the target stimulus may be impaired if it appears immediately after the preceding one [57,58]. From a neural perspective, attentional blink involves the right temporoparietal junction (rTPJ) and the left inferior frontal junction (lIFJ) [59].

The right hemisphere plays a particularly significant role in attentional blink. Studies on split-brain patients have shown that when the second target was presented to the right hemisphere, attentional blink lasted significantly longer, indicating the key role of the right hemisphere in processing visual information under limited capacity conditions [60]. In patients with right-hemisphere stroke, the effective field of vision is reduced, and the attentional blink interval is prolonged, particularly for left-sided stimuli [61]. Additionally, transcranial magnetic stimulation (TMS) of the right posterior parietal cortex reduces the magnitude of attentional blink, further confirming the critical role of the right hemisphere in the temporal aspects of visual attention [62].

The right hemisphere is critically important for multitasking in various contexts. Studies have shown that lesions in the right temporoparietal region can lead to chronic multitasking impairments, even after the initial deficits, such as neglect, have been rehabilitated [63]. Neuroimaging research has identified a predominantly right-lateralized fronto-parietal network and the cerebellum as key regions activated during dual-task performance [64]. A meta-analysis has revealed a multitasking-related neural network that includes the bilateral intraparietal sulcus, left dorsal premotor cortex, and right anterior insula [65]. However, multitasking and task switching have distinct neural correlates [64–66]: task switching activates the left premotor and inferior parietal areas, whereas multitasking engages the right prefrontal and inferior parietal cortex.

Thus, the limitations of simultaneous and successive cognitive task performance are natural even in a healthy brain, but brain pathology exacerbates the depletion of cognitive resources. It is suggested that cognitive reserve helps protect against the consequences of brain damage. However, the effectiveness of cognitive reserve decreases under multitasking conditions, exposing a "gray zone" between hidden and overt behavioral deficits [67].

The dual-task paradigm became a modern approach to assess spatial attention deficits in neglect patients by combining two simultaneous tasks, typically a visual task with an auditory or cognitive task. For instance, peripheral target detection can be combined with central shape recognition [68], digit indication [69] or lane tracking in a driving simulator [70]. Other designs include visual tasks with concurrent non-spatial or spatial auditory tasks [71]. Studies have demonstrated that dual-task conditions, which increase attentional load, can exacerbate VN symptoms and reveal subtle, well-compensated forms of dysfunction [72,73].

Computerized dual-task assessments lead to more contralesional omissions [73,74] and slower contralesional reaction times [75] than single-task conditions. Additionally, large-screen dual-task presentations have been effective in detecting subclinical forms of VN [72]. VN symptoms that remain hidden during single-task conditions may emerge under multitasking demands [67]. Importantly, dual-task paradigms have revealed that not only right hemisphere damage can result in contralesional deficits: Blini et al. [76] demonstrated that patients with left hemisphere strokes, who showed no signs of neglect on standard tests, exhibited significant right-sided omissions under dual-task conditions. This challenges the traditionally unilateral view of neglect and supports the idea that supramodal attentional resources—rather than strictly spatial mechanisms—are taxed during multitasking, potentially unmasking deficits in either hemifield. In contrast, studies with healthy participants [77,78] show that although increased attentional load can modulate perceptual integration (e.g., enhancing the sound-induced flash illusion), it does not produce lateralized effects. This reinforces the interpretation that spatial asymmetries under cognitive load reflect pathological processing. Taken together, this body of evidence supports the utility of dual-task paradigms not only for diagnosing subtle or compensated forms of neglect but also for probing broader attentional vulnerabilities following focal brain injury.

Dual-task conditions are not only useful for diagnosing VN but also for its rehabilitation. While some studies indicate that dual-task training improves gait, cognitive processes, and skill transfer across various neurological disorders [79], others have found no significant additional benefits over single-task training in neglect rehabilitation [80]. The effectiveness of multitasking-based interventions may depend on factors such as training duration, task complexity, and individual patient characteristics. Despite mixed findings, dual-task paradigms remain a promising therapeutic approach in neurological rehabilitation, with the potential to target deficits in both spatial and non-spatial attention [79,80].

In this study, we developed the Keen Eye method using a computer-based dual-task paradigm, building upon a well-established tradition of using increased cognitive load to reveal subtle or compensated forms of VN. The primary goal of this approach is to surpass the diagnostic sensitivity of standard paper-and-pencil tests, identifying even subclinical or residual forms of neglect that often go undetected in conventional assessments. Dual-task paradigms in spatial neglect research are known to measure how divided attention and limited cognitive capacity affect spatial awareness. Accordingly, we formulated the following hypotheses. First, patients with visual neglect would exhibit significantly more omissions of targets presented in the contralesional (left) hemispace under dual-task conditions, compared to patients without neglect. Second, we expected that the Keen Eye method would demonstrate sufficient sensitivity to detect signs of neglect even in individuals who did not meet conventional diagnostic criteria based on behavioral observation or standard paper-and-pencil tests. Third, due to the vertical arrangement of target positions, we hypothesized that the method would allow for a more nuanced, dimensional assessment of neglect, with a particular emphasis on omissions in the lower-left quadrant, where spatial deficits are often most pronounced.

## Materials and methods

### Participants

The study sample included 102 participants with right hemisphere damage, of whom 38 were diagnosed with visual neglect (VN) and 64 were patients without neglect. The average age of the patients with neglect was 65.6 ± 10.7 years (minimum 28, maximum 80), and the average age of those without neglect was 60.9 ± 12.3 years (minimum 29, maximum

82). Among the participants, there were 46 women and 57 men. The study employed prospective sampling as patients were admitted to the neurology department. All participants had documented right hemisphere damage confirmed by neuroimaging (CT scans). All patients were in the subacute or chronic phase of recovery; no individuals in the acute phase (within the first month post-injury) were included. Time since stroke or injury ranged from 1 to 16 months ($5.86 \pm 4.58$ months). The majority of patients (n = 79; 77%) experienced ischemic strokes, while 21 patients (21%) had hemorrhagic strokes. Additionally, 2 participants (2%) had right-hemisphere traumatic brain injuries (TBI), with documented damage to the parietal or temporo-parieto-occipital areas. Most vascular lesions were localized in the territory of the right middle cerebral artery (MCA), with several cases involving the posterior cerebral artery (PCA), anterior cerebral artery (ACA), thalamic regions, or displaying widespread right-hemisphere involvement. Generalized clinical characteristics for each study participant are provided in Supplementary File S1 File.

The diagnosis of visual neglect was established by a board-certified neurologist based on a comprehensive neurological and neuropsychological assessment of higher mental functions [81]. Clinical criteria included (i) clear signs of inattention to stimuli in the left visual field, (ii) anosognosia for the deficit, and (iii) neuropsychological assessment. A comprehensive neuropsychological assessment was conducted by an experienced neuropsychologist to identify signs of visual neglect. This assessment combined standardized quantitative tests with qualitative methods adapted from A.R. Luria's classical neuropsychological approach [81].

Quantitative tests:

- Albert's Test [82]: A line cancellation task; ≥ 2 omissions on the left side were considered indicative of spatial inattention;

- Bells Test [29]: A symbol cancellation test; ≥ 6 omissions on the left side supported evidence of neglect;

- Catherine Bergego Scale (CBS): An observational checklist evaluating neglect symptoms in daily functioning to increase ecological validity (0 = No behavioral neglect, 1–10 = Mild behavioral neglect, 11–20 = Moderate behavioral neglect, 21–30 = Severe behavioral neglect [83]).

Qualitative assessment, based on Lurian methods [81]:

- Five Figures Task: Reproduction of a series of five geometric figures from memory; typical indicators included left-side omissions, rightward bias in figure placement, and initiating drawing from the right.

- Table Drawing Task: Patients drew a simple table; signs of neglect included incomplete rendering of the left side of the table.

- Clock Drawing Test: Patients sketched a clock face with numbers and hands; signs included placing numbers and hands only on the right side, omitting left-side numerals.

Throughout testing, special attention was given to patients' scanning strategies (e.g., starting search on the right), midline reading shifts, naming limited to the right side, and awareness of omissions. This neuropsychological approach ensured a robust identification of neglect features. All participants were assessed using the complete set of described neuropsychological tests to the extent possible given their clinical condition. Detailed scoring protocols and illustrative examples are available in Supplementary File S2 File.

Exclusion criteria included significant speech or perceptual disorders, disorientation, and confusion. Patients in affective states that interfered with interaction with the psychologist were also excluded.

The study followed the ethical principles of the Helsinki Declaration. This study conforms to ethical standards and was approved by The Ethics Committee of The Federal State Budget Scientific Institution «Federal Scientific Center of Psychological and Multidisciplinary Research» (Approval data: 16.05.2024 (№5)). All patients were informed in advance about the purpose of this study and signed informed consent forms.

## Materials

To carry out the Keen Eye method, a laptop with a screen size of at least 28 cm and a wireless mouse with a mouse-pad were required. PsychoPy software was used in the study to conduct Keen Eye method. The study was conducted primarily on Huawei MateBook 14 laptops with a 15.6-inch screen running Windows 11. The method's functionality was also tested on other operating systems, including Linux (with a recommendation to check the system locale for correct data storage if the system language is English) and macOS, as well as on laptops with similar screen sizes and technical specifications.

## Procedure of one trial

The main task for the participant is to determine the location of a circle with a diameter of 8 mm. Simultaneously, the participant performs the secondary task of identifying the central digit (1, 2, 3, or 4). This task combination was chosen because the central digit task competes for attentional resources with peripheral stimulus detection, thus mimicking real-life multitasking demands. Each trial starts with the presentation of a fixation cross for 1000 ms. Then, a circle appears on the periphery, and a digit appears at the center of the screen for 100 ms. Afterward, a black-and-white mask (in the form of a randomly generated dot diagram) appears on the screen for 2000 ms (Fig 1). At the end of one trial, the participant orally reports to the experimenter which digit was in the center of the screen and on which side the circle was located. After the mask, a response input screen appears, where the experimenter enters the participant's verbal report about what was seen on the screen.

Based on previous findings showing that patients with visual neglect typically exhibit significantly fewer omissions in single-task conditions than in dual-task paradigms [69,73,74], we decided not to include a single-task condition in the

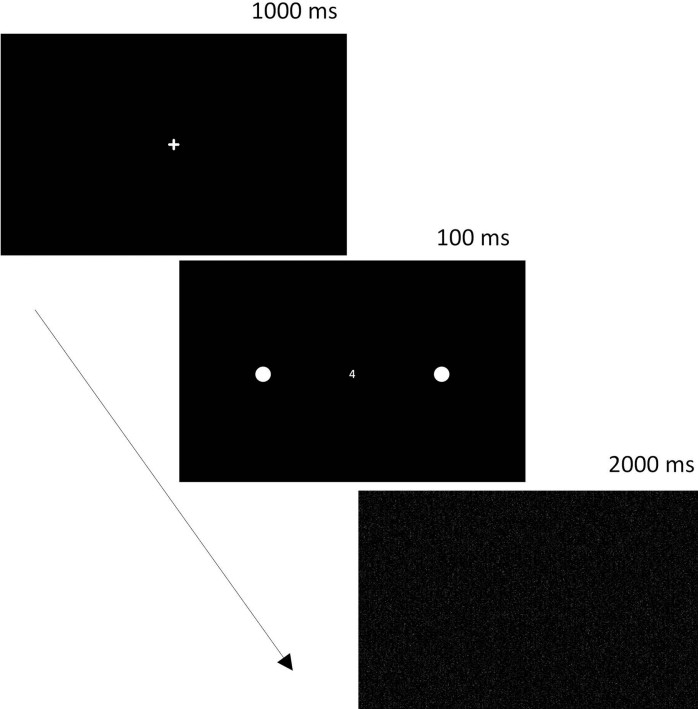

**Fig 1. Sequence of events in a single trial.**

current protocol. Including both would have substantially increased the total number of trials, which could lead to fatigue and reduced compliance in a clinical population. Instead, we focused solely on dual-task conditions, which are known to be more sensitive to subtle attentional deficits. This allowed us to reduce testing time and cognitive load while maintaining sufficient spatial coverage across multiple visual field locations. Our priority was to balance diagnostic sensitivity with ecological feasibility for patients in clinical settings.

### Instruction

The participant is given the following standard instruction:

> "Watch closely what happens on the screen. First, a cross will appear in the center of the screen; focus on it. Then a digit will appear at the cross's location. Simultaneously with the digit, a circle will appear – either on the right, left, or both sides. The circle can appear in different parts of the screen. Be attentive: the digit and the circle will appear for a very short time! Try to see them. Then say aloud what you managed to notice: the digit and the side where the circle was, for example, 'One on the right.'"

### Trial parameters

Before the experiment begins, a screen displaying an example of how the fixation cross, digit, and circle (placed 13.5 cm to the right of the center) will appear is shown. This allows the participants to become familiar with the stimuli and their sizes, and the experimenter can check the settings to ensure proper display of distances on the specific device.

At the beginning of the experiment, participants complete a training series consisting of 7 trials. During the training, participants familiarize themselves with the possible trial variations, where parameters such as the number of circles (1 or 2), presentation side, and distance from the screen's center (both vertically and horizontally) are varied. The stimulus presentation time decreases from 1 second to 100 ms during the training series.

Following the training, the participant proceeds to the main series. The main series consists of 75 trials: 42 unilateral presentations (21 with a circle on the right side of the center and 21 with a circle on the left side) and 33 bilateral presentations. The number of trials per stimulus position was determined based on methodological principles aiming to maximize measurement reliability while minimizing participant fatigue. Each of the 25 target configurations (including both single and bilateral presentations) was shown three times, ensuring that detection failures were not due to random attentional lapses but reflected consistent spatial biases. This repetition rate was refined through pilot testing, which confirmed that three presentations per configuration yielded stable omission rates without causing excessive fatigue in clinical participants. Regarding spatial layout, target eccentricities were designed to cover a range from foveal to peripheral regions: positions closest to the center subtended approximately 1.4 degrees of visual angle, while the most eccentric positions extended up to 13.2 degrees. This distribution allowed us to test the influence of both central and peripheral spatial locations on neglect-related omissions, thereby enhancing the ecological validity and diagnostic sensitivity of the method. Overall, the total number of trials was chosen to ensure sufficient statistical power for within-subject and between-group analyses while keeping the task duration acceptable for patients with right hemisphere lesions, who are particularly susceptible to cognitive fatigue. All trials are presented in a randomized order. There are 7 possible positions for the circle on each side, resulting in 14 possible circle positions on the screen (Fig 2).

The distance at which each of the 14 circles is located from the center of the screen is presented in Table 1. Participants were seated at distance of approximately 60 cm from the screen. Although the use of a chinrest was initially planned to standardize viewing distance, it was ultimately avoided because many patients, due to age or clinical condition, found it uncomfortable or refused its use. Instead, the experimenter ensured that a consistent distance was maintained by providing clear verbal instructions and monitoring the patient's position throughout the task. At this distance, the target positions corresponded

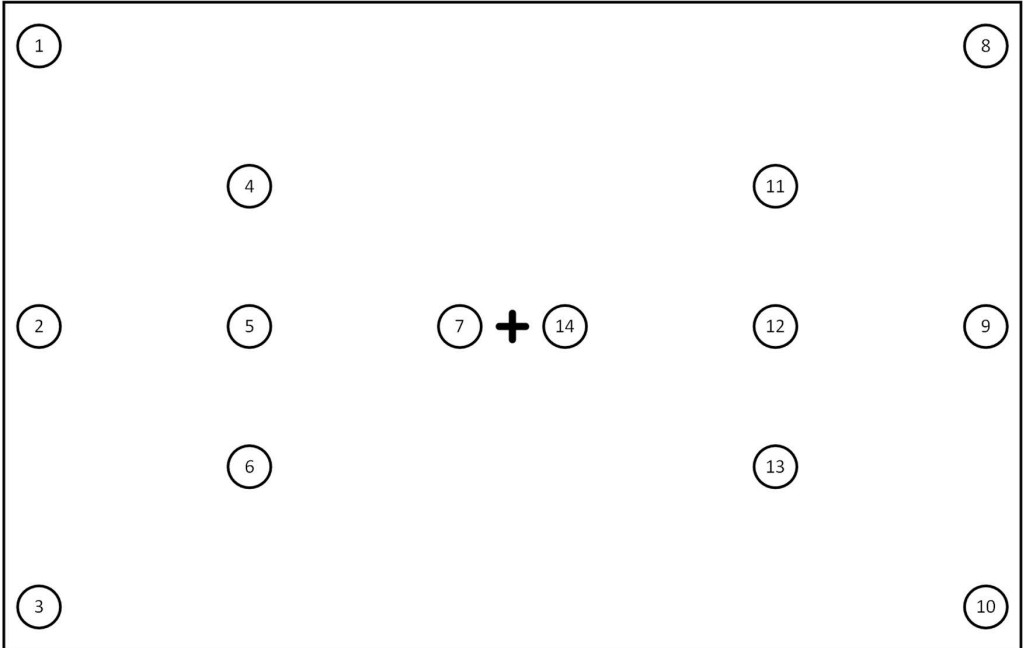

**Fig 2. Possible positions of the target stimulus.**

**Table 1. Circle positions relative to the center of the screen.**

| Target position number | Horizontal distance from the center of the screen (cm) | Vertical distance from the center of the screen (cm) |
|---|---|---|
| 1 | −13.5 | 8 |
| 2 | −13.5 | 0 |
| 3 | −13.5 | −8 |
| 4 | −7.5 | 4 |
| 5 | −7.5 | 0 |
| 6 | −7.5 | −4 |
| 7 | −1.5 | 0 |
| 8 | 13.5 | 8 |
| 9 | 13.5 | 0 |
| 10 | 13.5 | −8 |
| 11 | 7.5 | 4 |
| 12 | 7.5 | 0 |
| 13 | 7.5 | −4 |
| 14 | 1.5 | 0 |

to an approximate visual angle of 1.4°–13.2° horizontally and 1.9°–7.6° vertically, depending on each stimulus's eccentricity. Task parameters were defined in centimeters relative to the screen center for clarity and reproducibility, but the corresponding degrees of visual angle have now been added to account for variability in screen size and viewing distance. This information ensures methodological transparency and supports replicability in different research and clinical contexts.

Table 2 and Fig 3 provide characteristics of all 75 trials in the main series.

**Table 2. All possible combinations presented in the method.**

| Type of exposure | | Target position numbers | Number of exposures |
|---|---|---|---|
| **Unilateral** | | 1 | 3 |
| | | 2 | 3 |
| | | 3 | 3 |
| | | 4 | 3 |
| | | 5 | 3 |
| | | 6 | 3 |
| | | 7 | 3 |
| | | 8 | 3 |
| | | 9 | 3 |
| | | 10 | 3 |
| | | 11 | 3 |
| | | 12 | 3 |
| | | 13 | 3 |
| | | 14 | 3 |
| **Bilateral** | **Symmetrical to the vertical axis** | 1 + 8 | 3 |
| | | 2 + 9 | 3 |
| | | 3 + 10 | 3 |
| | | 4 + 11 | 3 |
| | | 5 + 12 | 3 |
| | | 6 + 13 | 3 |
| | | 7 + 14 | 3 |
| | **Symmetrical to the diagonal axis** | 1 + 10 | 3 |
| | | 3 + 8 | 3 |
| | | 4 + 13 | 3 |
| | | 6 + 11 | 3 |

## Data processing in the Keen Eye method

Based on the analysis of the method, the following variables are computed:

- 1–14 – number of errors in the corresponding sectors of the screen (see Fig 2);

- right_skipped_unilateral (RSU) – number of missed right stimuli in the unilateral presentation;

- left_skipped_unilateral (LSU) – number of missed left stimuli in the unilateral presentation;

- right_skipped_bilateral (RSB) – number of missed right stimuli in the bilateral presentation;

- left_skipped_bilateral (LSB) – number of missed left stimuli in the bilateral presentation; right_hallucination (RH) – number of responses indicating "right" when there was no circle on the right side;

- left_hallucination (LH) – number of responses indicating "left" when there was no circle on the left side;

- number_skipped_right (NIR) – number of errors in identifying central digit when a circle was presented on the right side;

- number_skipped_left (NIL) – number of errors in identifying central digit when a circle was presented on the left side;

- number_skipped_bilateral (NIB) – number of errors in identifying central digit when circles were bilaterally presented An incorrect response was defined as either misidentifying the digit or failing to provide a definite answer (e.g., responding

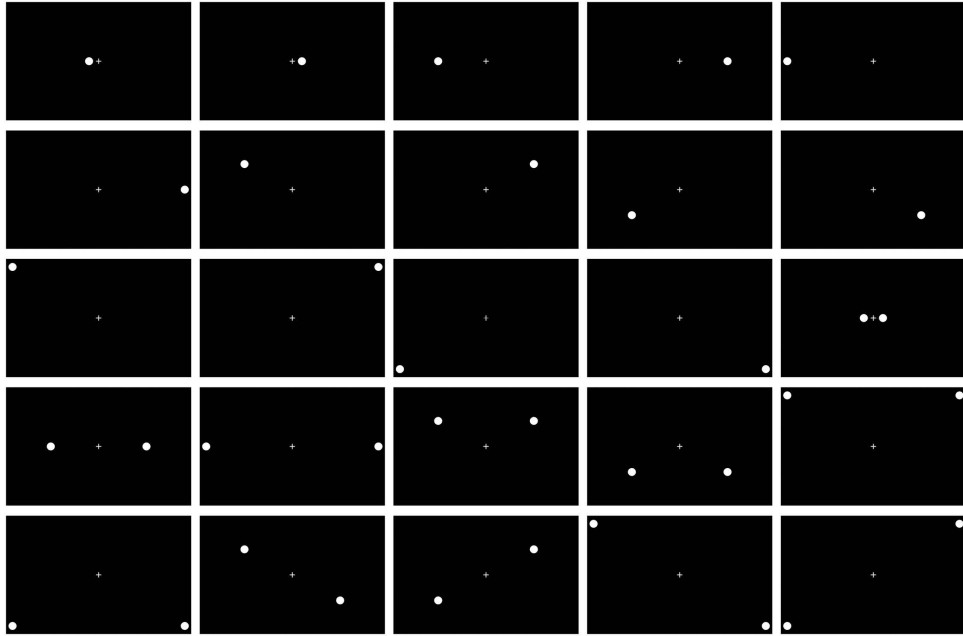

**Fig 3. All possible combinations of circles presented in the method.**

"I didn't see" or "something flashed"). All such cases were counted as errors in the digit task. This variable was treated as an index of additional attentional load and not analyzed in detail, since the primary aim of the study was to assess spatial neglect.

Individual target positions were introduced to provide a more nuanced and individualized assessment of attentional deficits within specific regions of space, particularly for detecting neglect along both the horizontal and vertical axes. Separate indices for omissions of unilateral and bilateral target presentations were included to differentiate between single-target neglect and extinction occurring under multiple-target conditions. The LH and RH indices were designed to evaluate optical allochiria, while error counts on the central task served to assess non-lateralized attentional deficits. In addition, incorrect identifications of the central digit at different spatial target positions could, in future research, be analyzed separately to explore how spatial load influences performance on the central task.

Additionally, similar to dichotic listening [84], the following coefficients are calculated:

- KA_bilateral – asymmetry coefficient in the bilateral presentation;

- KPrR_bilateral – right productivity coefficient in the bilateral presentation;

- KPrL_bilateral – left productivity coefficient in the bilateral presentation;

- KA_all – general asymmetry coefficient;

- KEf_all – general efficiency coefficient;

- KPrR_all – overall right productivity coefficient;

- KPrL_all – overall left productivity coefficient.

The asymmetry coefficients are calculated by the program using the following formulas:

**KA_bilateral** – asymmetry coefficient in the bilateral presentation. This coefficient ranges from −1 to +1 and allows the assessment of the degree of neglect during bilateral stimulus perception. It also indicates whether there is a spatial bias toward the ipsilesional or contralesional hemispace during bilateral presentations. In the absence of a lateralized attention deficit, this parameter should approach 0. It is calculated as:

$$KA_{bilateral} = \frac{LSB - RSB}{2BP - RSB - LSB}$$

where *BP* equals 33, indicating the number of bilateral presentations.

**KPrR_bilateral** – right productivity coefficient in the bilateral presentation of circles. This describes the proportion of "caught" circles on the right during bilateral presentations and ranges from 0 to 1. It reflects the share of correct detections relative to all right-sided target presentations, regardless of their total number. It is calculated as:

$$KPrR_{bilateral} = \frac{BP - RSB}{BP}$$

**KPrL_bilateral** – left productivity coefficient in the bilateral presentation of circles. This describes the proportion of "caught" circles on the left during bilateral presentations and ranges from 0 to 1. It is calculated as:

$$KPrL_{bilateral} = \frac{BP - LSB}{BP}$$

**KA_all** – overall asymmetry coefficient. This coefficient ranges from −1 to +1 and allows the assessment of neglect during both bilateral and unilateral stimulus perceptions. It represents the broadest index of ipsilesional or contralesional bias and the severity of this bias. In the absence of a lateralized attention deficit, this parameter should approach 0. It is calculated as:

$$KA_{all} = \frac{LSB + LSU - RSB - RSU}{2LP - RSB - RSU - LSB - LSU}$$

where LP equals 54, indicating the total number of lateralized (left and right) circle presentations.

**KEf_all** – overall efficiency coefficient. This coefficient ranges from 0 to 1 and accounts for spatial mislocalization errors (e.g., allocheria). By quantifying incorrect left–right spatial assumptions, it allows the detection of patients who have difficulties in determining allocentric coordinates and the severity of such errors. It is calculated as:

$$KEf_{all} = \frac{2LP - RSB - RSU - LSB - LSU - RH - LH}{2LP - RSB - RSU - LSB - LSU + RH + LH}$$

**KPrR_all** – overall right productivity coefficient. This describes the proportion of "caught" circles on the right during all circle presentations and ranges from 0 to 1. It is calculated as:

$$KPrR_{all} = \frac{LP - RSB - RSU}{LP}$$

**KPrL_all** – overall left productivity coefficient. This describes the proportion of "caught" circles on the left during all circle presentations and ranges from 0 to 1. It is calculated as:

$$KPrL_{all} = \frac{LP - LSB - LSU}{LP}$$

For the justification of convergent validity, Albert's Test [82] and The Bells Test [29] were selected. These standardized methods were chosen because they are widely recognized techniques used for the quantitative and qualitative diagnosis of patients with unilateral spatial neglect in the visual modality.

## Statistical analysis

All statistical analyses were performed using RStudio 2023.12.1 Build 402 (packages: rattle, rpart, rpart.plot) and jamovi (version 2.3.18). Descriptive statistics, including mean values, standard deviations, medians, and interquartile ranges, were calculated for all variables. The Mann-Whitney U test was used for non-parametric data. The Kruskal-Wallis test was used for multiple group comparisons, and the post hoc DSCF test with Holm's correction was applied for multiple comparisons.

The effect size was calculated using rank-biserial correlation for non-parametric tests. To assess the relationship between the severity of neglect and frontal dysfunction, a correlation analysis was performed. Cluster analysis was conducted in jamovi using k-means clustering with silhouette analysis to determine the optimal number of clusters.

## Results

### Comparison between groups on the Keen Eye method indicators

Descriptive statistics were calculated for all sample parameters. Table 3 presents the mean (M), standard deviation (SD), minimum, and maximum values for each variable in the patient group with VN and the patient group without neglect. Figs 4-6 summarize the distributions of core methodology parameters through boxplot representations. Figs 4 and 5 display the percentage of omissions relative to the maximum possible number of target presentations. The results indicate that the average number of omissions on the left side is consistently higher in patients with neglect, with some individuals reaching the maximum possible number of errors on specific measures. Additionally, in patients with VN, the minimum number of errors for certain parameters (e.g., LSB) is notably distant from zero, suggesting variability in performance even among affected individuals.

Figs 4–6 clearly demonstrate a contralesional attentional bias – that is, a rightward asymmetry of visual attention when the spatial task is performed under dual-task conditions. As shown in Fig 4, both patients with neglect and right-hemisphere patients without visual neglect make more omissions for targets presented on the left side. This asymmetry is substantially greater in patients with neglect, as confirmed by within-group comparisons of total omissions on the right (RSU + RSB) versus the left (LSU + LSB). For the neglect group, the Wilcoxon test yielded W = 741, p < 0.001 ($r_{rb}$ = 1); for the non-neglect group, W = 1116, p = 0.001 ($r_{rb}$ = 0.502).

The magnitude of this spatial bias is also supported by the indices shown in Fig 6. The KA_bilateral and KA_all coefficients quantify spatial attention asymmetry: values approaching −1 indicate ipsilesional bias, values approaching +1 indicate contralesional bias, and values near 0 reflect no asymmetry. As illustrated in Fig 6, the group of patients without neglect shows almost no bias, whereas patients with neglect exhibit coefficients close to +1. This pattern is expected for neglect patients and aligns with the findings of most previous studies.

A comparison was made between groups of right hemisphere patients without neglect and those with neglect using the Mann-Whitney U test. The results of the analysis are presented in Table 4. After applying Holm's correction for multiple comparisons, significant group differences (p < 0.05) were observed for most indicators, including key omission measures (LSU, LSB) and major asymmetry coefficients (KA_all, KA_bilateral, KPrL_bilateral), as well as several derived indices (NIR, NIL, NIB). No significant differences were found for LH, RSB, KPrR_all, or KPrR_bilateral after correction. Among omission error sums, LSB and LSU remained the strongest indicators, underscoring the specificity of this paradigm for detecting left-sided neglect. Asymmetry coefficients such as KA_all and KA_bilateral provide an ordinal index of neglect severity ranging from −1–1 and are not dependent on the absolute number of targets.

### Diagnostic criterion selection using decision trees

To identify the most significant diagnostic criteria for neglect using the Keen Eye method, decision tree analyses were conducted. This approach allows data classification based on features selected from the original dataset, with the goal of building a predictive model for diagnosis using specified predictor variables.

**Table 3. Descriptive statistics for the groups of patients with and without visual neglect.**

| Variable | Without VN Group | | VN group | |
|---|---|---|---|---|
| | (M ± SD) | Min – Max | (M ± SD) | Min – Max |
| 1 | 3.578 ± 3.554 | 0 - 9 | 6.132 ± 3.488 | 0 - 9 |
| 2 | 2.328 ± 2.482 | 0 - 6 | 4.131 ± 2.244 | 0 - 6 |
| 3 | 3.828 ± 3.623 | 0 - 9 | 6.236 ± 3.693 | 0 - 9 |
| 4 | 2.687 ± 3.279 | 0 - 9 | 6.131 ± 3.371 | 0 - 9 |
| 5 | 1.765 ± 2.180 | 0 - 6 | 3.921 ± 2.432 | 0 - 6 |
| 6 | 2.921 ± 3.296 | 0 - 9 | 6.236 ± 3.590 | 0 - 9 |
| 7 | 1.421 ± 1.875 | 0 - 6 | 3.421 ± 2.606 | 0 - 9 |
| 14 | 0.750 ± 1.195 | 0 - 5 | 1.211 ± 1.597 | 0 - 6 |
| 11 | 0.859 ± 1.531 | 0 - 7 | 0.868 ± 1.418 | 0 - 6 |
| 12 | 0.563 ± 0.889 | 0 - 4 | 0.947 ± 1.355 | 0 - 5 |
| 13 | 1.000 ± 1.448 | 0 - 6 | 1.921 ± 2.353 | 0 - 8 |
| 8 | 1.344 ± 1.986 | 0 - 9 | 1.921 ± 2.465 | 0 - 9 |
| 9 | 0.703 ± 1.365 | 0 - 9 | 1.026 ± 1.636 | 0 - 6 |
| 10 | 1.531 ± 2.225 | 0 - 9 | 2.395 ± 2.881 | 0 - 9 |
| RSU | 1.922 ± 2.978 | 0 - 14 | 4.289 ± 4.299 | 0 - 16 |
| LSU | 4.656 ± 5.782 | 0 - 21 | 16.368 ± 5.749 | 1 - 21 |
| RSB | 4.531 ± 6.510 | 0 - 32 | 6.500 ± 6.479 | 0 - 27 |
| LSB | 7.953 ± 10.045 | 0 - 33 | 29.815 ± 4.980 | 14 - 33 |
| NIR | 3.984 ± 4.971 | 0 - 21 | 10.500 ± 8.013 | 0 - 21 |
| NIL | 3.609 ± 4.699 | 0 - 21 | 9.816 ± 7.651 | 0 - 21 |
| NIB | 5.250 ± 6.785 | 0 - 33 | 12.421 ± 11.337 | 0 - 33 |
| RH | 0.391 ± 0.866 | 0 - 5 | 1.632 ± 3.062 | 0 - 16 |
| LH | 0.125 ± 0.333 | 0 - 1 | 0.132 ± 0.343 | 0 - 1 |
| KA_bilateral | 0.101 ± 0.324 | −0.941 - 1 | 0.826 ± 0.239 | 0.136 - 1 |
| KPrR_bilateral | 0.863 ± 0.197 | 0.030 - 1 | 0.803 ± 0.196 | 0.182 - 1 |
| KPrL_bilateral | 0.759 ± 0.304 | 0 - 1 | 0.097 ± 0.151 | 0 - 0.576 |
| KA_all | 0.104 ± 0.274 | −0.479 - 0.927 | 0.746 ± 0.247 | 0.182 - 1 |
| KEf_all | 0.987 ± 0.033 | 0.760 - 1 | 0.927 ± 0.126 | 0.407 - 1 |
| KPrR_all | 0.881 ± 0.164 | 0.278 - 1 | 0.800 ± 0.197 | 0.204 - 1 |
| KPrL_all | 0.767 ± 0.285 | 0.037 - 1 | 0.145 ± 0.172 | 0 - 0.667 |

Positions of individual targets are ordered left-to-right and top-to-bottom (1–7, 14, 11–13, 8–10).

To evaluate the performance of the decision tree model, several statistical measures were calculated. The Gini index served as the splitting criterion for decision tree construction, measuring node impurity on a scale from 0 (indicating a pure node containing only one class) to 0.5 (representing maximum impurity in binary classification problems). Lower Gini index values indicate more effective class separation at each node. Model performance was assessed using standard classification metrics. Precision was calculated as the proportion of true positive predictions among all positive predictions, quantifying the accuracy of positive class predictions. Recall, also known as sensitivity, represented the proportion of actual positive cases that were correctly identified, measuring the model's ability to detect all positive instances. The F1 score was computed as the harmonic mean of precision and recall, providing a balanced performance measure that accounts for both false positives and false negatives. The F1 score is particularly valuable when both precision and recall are equally important for the classification task, as it penalizes models with large disparities between these two metrics.

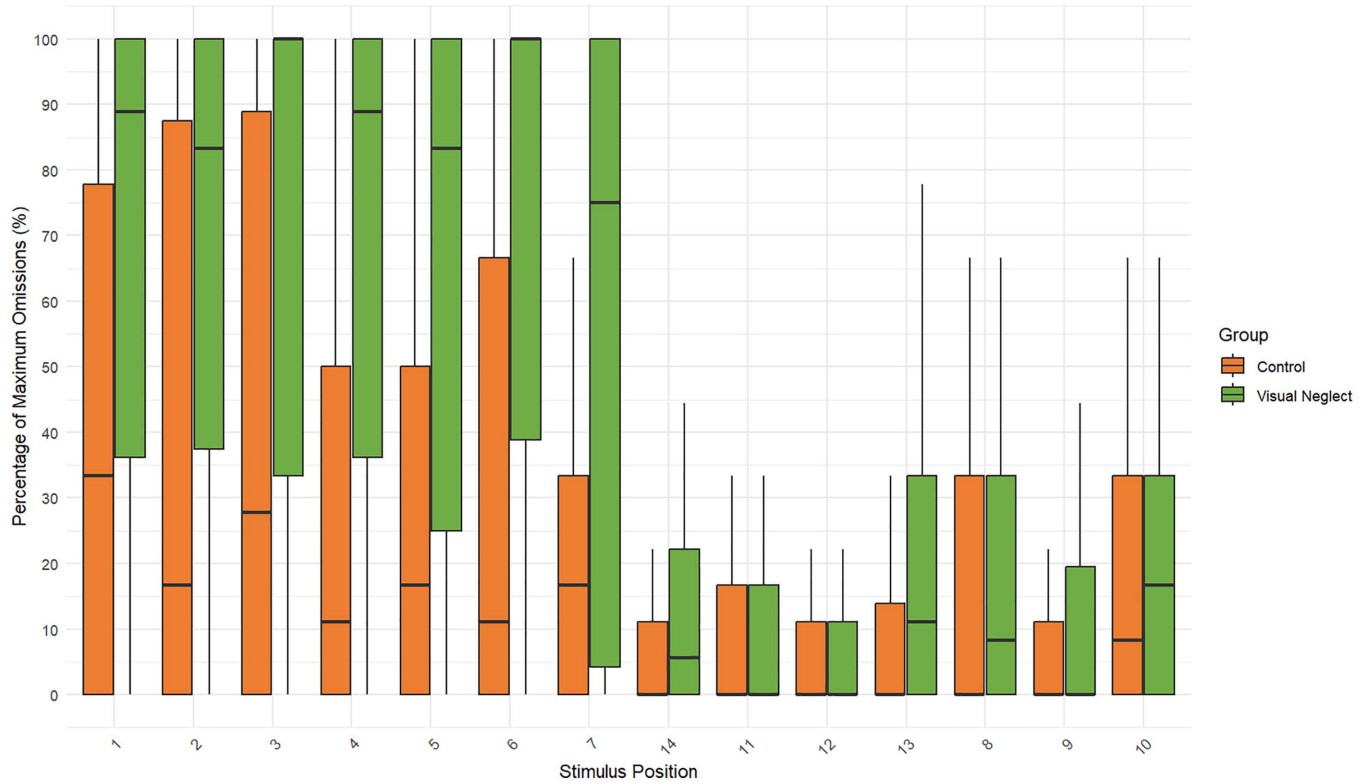

**Fig 4. Boxplot of omission percentages for each individual circle position in the Keen Eye method across both groups.** Positions are ordered left-to-right and top-to-bottom (1-7, 14, 11-13, 8-10). The y-axis represents the percentage of maximum possible omissions (%), i.e., the proportion of missed targets normalized by the maximum number of presentations for each position (positions 2, 5, 7, 9, 12, and 14 were presented 6 times; all other positions 9 times). Normalization was applied to allow valid comparison across positions with different presentation frequencies. The black horizontal line inside each box indicates the median, and dots represent outliers.

The predictors for classification models 1 and 2 included total error counts (RSU, LSU, RSB, LSB, RH, and LH). Model 1 (Fig 7) was constructed using cross-validation (Accuracy = 0.90, Precision = 0.82, Recall = 0.95, F1 Score = 0.88, Sensitivity = 0.95, Specificity = 0.88). The decision tree model was constructed using 10-fold cross-validation to ensure robust performance estimation and prevent overfitting. Cross-validation was implemented through the rpart.control function with the xval parameter set to 10, creating 10 equal-sized folds from the dataset. To maintain model stability and interpretability, several tree-growing parameters were specified. The minimum split parameter (minsplit) was set to 20, requiring at least 20 observations to be present in a node before attempting a split, thereby preventing the creation of overly specific decision rules based on small sample sizes. The minimum bucket parameter (minbucket) was established at 7, ensuring that each terminal node contained a minimum of 7 observations to maintain statistical reliability of the final classifications. Model complexity was controlled through the complexity parameter (cp) set to 0.01, which determines the minimum improvement in model fit required to justify additional tree splits. This parameter serves as a pruning threshold, preventing the model from becoming overly complex and reducing the risk of overfitting to the training data. These parameter settings collectively ensure that the resulting decision tree maintains an appropriate balance between model complexity and generalizability.

Model 2 (Fig 8) was designed to maximize recall and improve specificity (Accuracy = 0.87, Precision = 0.75, Recall = 1.00, F1 Score = 0.85, Sensitivity = 1.00, Specificity = 0.80).

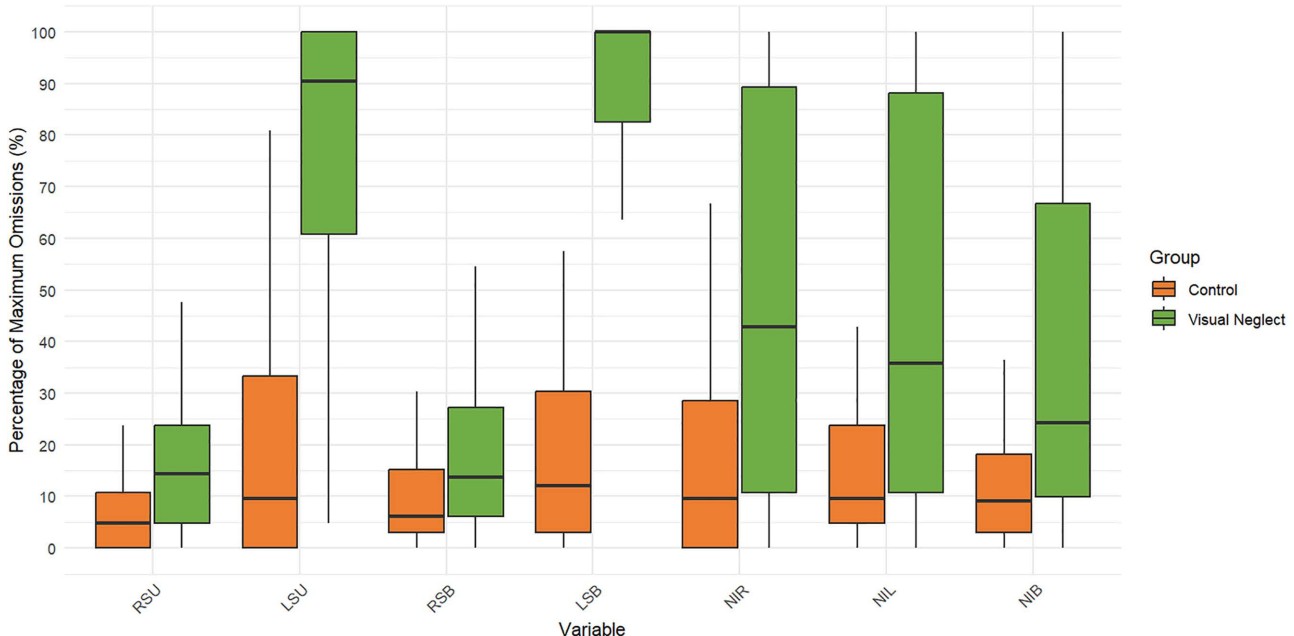

**Fig 5. Boxplot of total omission error percentages in the Keen Eye method for both groups.** The y-axis represents the percentage of maximum possible omissions (%), calculated as the total number of missed targets relative to the maximum possible number of target presentations. For RSU, LSU, NIR, and NIL the maximum was 21 presentations; for RSB, LSB, and NIB the maximum was 33 presentations.

Model 3 (Fig 9) was developed using predictors based on asymmetry coefficients and constructed using cross-validation (KA_bilateral, KPrR_bilateral, KPrL_bilateral, KA_all, KEf_all, KPrR_all, and KPrL_all) (Accuracy = 0.90, Precision = 0.79, Recall = 1.00, F1 Score = 0.88, Sensitivity = 1.00, Specificity = 0.84). The same cross-validation parameters were applied to model 3 as those used for model 1, maintaining consistency in the evaluation methodology across all models. However, we see that this model outperforms model 1 in terms of sensitivity and is comparable to model 2, since the predictors are coefficients that do not depend on the number of presentations.

All three models demonstrated high sensitivity; however, some patients without a diagnosed neglect condition were incorrectly classified into the neglect group. This may suggest that the Keen Eye method is capable of detecting hidden forms of VN, including well-compensated neglect.

From a diagnostic standpoint, the most important indicator is LSB. The LSB value is particularly useful for identifying visual extinction and can dissociate from LSU when a patient recognizes a single left-side presentation but fails to acknowledge the left side when two objects are presented in the perceptual field. When using Models 2 and 3 for diagnosing neglect, the method exhibits high specificity.

## Diagnosis of vertical neglect using the Keen Eye method

To identify vertical neglect, we selected 48 patients who, according to the Keen Eye method, were classified as "neglect" based on the criterion KA_all ≥ 0.17 (Model 3).

The following indicators were calculated to detect vertical neglect:

• up: the number of missed circles in positions above the horizontal line (positions 1, 4, 8, 11).

• down: the number of missed circles in positions below the horizontal line (positions 3, 6, 10, 13).

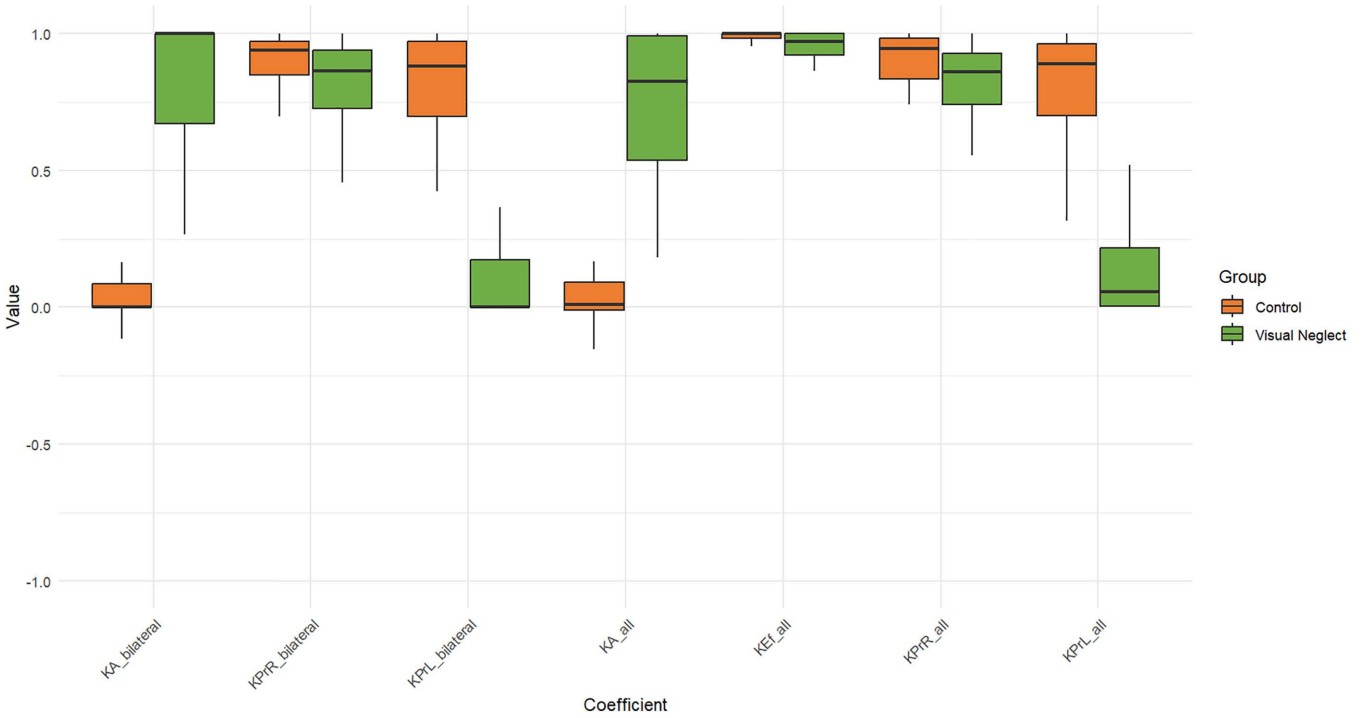

**Fig 6. Boxplot of asymmetry coefficients in the Keen Eye method for both groups.**

According to the literature, the left lower portion of the visual field tends to be neglected. Therefore, we used a directional hypothesis (down> up). The Wilcoxon signed-rank test was used to compare the related samples.

Results from the within-group comparison showed significant differences ($W = 663$, $p = 0.004$, $r_{rb} = 0.467$). The mean number of left-sided lower and upper omissions in the group of patients with neglect is presented in Fig 10.

Thus, the Keen Eye method aligns with findings of predominant neglect in the lower-left visual field in patients with neglect.

### Diagnostic efficiency of individual target positions

To evaluate the potential for reducing task length without compromising diagnostic accuracy, we analyzed the sensitivity and specificity of each of the 14 individual target positions. For this purpose, the quantitative variables (positions 1–14) were converted into binary form. The threshold for binarization was set at the median value for each variable, allowing us to divide the data into two groups: values greater than or equal to the median were encoded as 1, and values below the median were encoded as 0. Sensitivity and specificity metrics were then calculated for each position relative to the dichotomous variable diagnosis (Table 5).

Our analysis revealed that positions 4 and 6 demonstrated a relatively favorable balance between sensitivity and specificity. Specifically, position 4 showed the highest sensitivity (0.816) and relatively high specificity (0.656). Position 6 also had a high sensitivity (0.763) and the same specificity (0.656). In contrast, positions 14, 11, 12, and 9, despite having perfect sensitivity (1.0), exhibited zero specificity, indicating a high false-positive rate and limited diagnostic precision. Positions 5, 7, and 10 showed moderate sensitivity and specificity, but their diagnostic value is less pronounced compared to positions 4 and 6. The left eccentric positions (e.g., 1, 2, 3) demonstrated lower specificity than positions 4 and 6, which may limit their utility in precise diagnostics. However, eliminating diagonal or eccentric positions may significantly reduce

**Table 4. Intergroup comparisons of all parameters using the mann-whitney test and rank-biserial correlation.**

| Variable | U | p (raw) | Holm threshold | $r_{rb}$ |
|---|---|---|---|---|
| 1 | 729 | <.001 | 0.002 | 0.401 |
| 2 | 758 | 0.001 | 0.003 | 0.377 |
| 3 | 773 | 0.002 | 0.004 | 0.364 |
| 4 | 581 | <.001 | 0.003 | 0.522 |
| 5 | 686 | <.001 | 0.003 | 0.436 |
| 6 | 632 | <.001 | 0.003 | 0.481 |
| 7 | 706 | <.001 | 0.003 | 0.419 |
| 8 | 1082 | 0.325 | 0.013 | 0.111 |
| 9 | 1115 | 0.417 | 0.017 | 0.083 |
| 10 | 999 | 0.117 | 0.007 | 0.178 |
| 11 | 1178 | 0.77 | 0.025 | 0.031 |
| 12 | 1047 | 0.185 | 0.010 | 0.139 |
| 13 | 957 | 0.057 | 0.006 | 0.213 |
| 14 | 1033 | 0.16 | 0.008 | 0.150 |
| RSU | 710 | <.001 | 0.003 | 0.417 |
| LSU | 241 | <.001 | 0.002 | 0.802 |
| RSB | 890 | 0.023 | 0.006 | 0.269 |
| LSB | 153 | <.001 | 0.002 | 0.875 |
| NIR | 643 | <.001 | 0.002 | 0.471 |
| NIL | 618 | <.001 | 0.002 | 0.492 |
| NIB | 695 | <.001 | 0.002 | 0.428 |
| RH | 867 | 0.004 | 0.004 | 0.287 |
| LH | 1208 | 0.928 | 0.050 | 0.007 |
| KA_bilateral | 148 | <.001 | 0.002 | 0.878 |
| KPrR_bilateral | 890 | 0.023 | 0.005 | −0.269 |
| KPrL_bilateral | 153 | <.001 | 0.002 | −0.875 |
| KA_all | 146 | <.001 | 0.002 | 0.880 |
| KEf_all | 811 | 0.001 | 0.002 | −0.333 |
| KPrR_all | 823 | 0.006 | 0.005 | −0.324 |
| KPrL_all | 145 | <.001 | 0.004 | −0.881 |

the method's capacity to detect vertical or quadrant-specific forms of neglect. According to research on the structure of the perceptual field [85], extreme right stimuli (positions 8, 9, 10) are perceived worse even by healthy participants. Consequently, their value as a diagnostic tool requires further experimental verification. Therefore, any attempt to shorten the procedure must be made with caution. Definitive conclusions regarding the diagnostic optimization of the method should be postponed until further data are collected from both healthy controls and individuals with left hemisphere lesions.

## Psychometric properties of the method

Convergent validity was assessed using Guilford's Phi coefficient. The calculation was based on a comparison between the neuropsychologist's diagnosis and the diagnosis derived from test performance. The Keen Eye method identified LSB ≥ 14 as the threshold for diagnosing neglect, as determined by the classification tree method. Neglect was also diagnosed based on performance in paper-and-pencil tests, including the Bells Test [29] and Albert's Test [82]. Table 6 presents the correlation between the Keen Eye results and traditional neglect assessments.

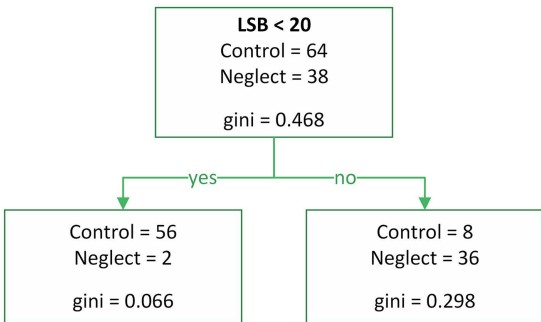

**Fig 7. Decision tree model 1 (Raw error counts, performance-optimized).** Model built using raw error indicators (RSU, LSU, RSB, LSB, RH, LH) and 10-fold cross-validation. Balanced performance: Sensitivity = 0.95, Specificity = 0.88.

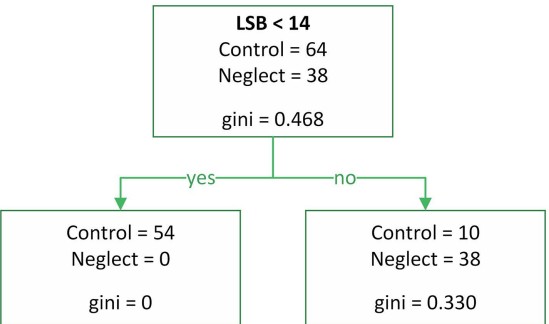

**Fig 8. Decision tree model 2 (Raw error counts, recall-maximized) Model built using raw error indicators (RSU, LSU, RSB, LSB, RH, LH), optimized to maximize recall and sensitivity.** Performance: Sensitivity = 1.00, Specificity = 0.80.

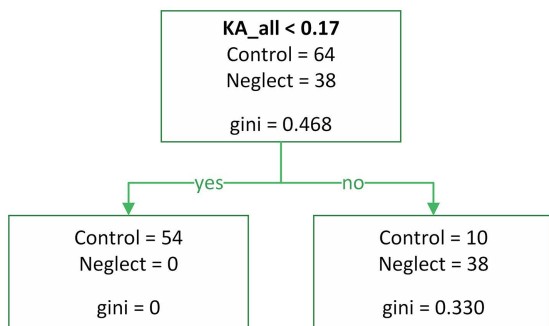

**Fig 9. Decision tree model 3 (Asymmetry coefficients).** Built using asymmetry and efficiency indices (KA_bilateral, KPrR_bilateral, KPrL_bilateral, KA_all, KEf_all, KPrR_all, and KPrL_all) and 10-fold cross-validation. Performance: Sensitivity = 1.00, Specificity = 0.84.

From the perspective of criterion validity, the Keen Eye method shows high scores, surpassing the diagnostic results of The Bell's Test [29] and Albert's Test [82], which also suggests high incremental and concurrent validity for the computer-based method. From the perspective of convergent validity, the Keen Eye test shows moderate similarity to The Bell's Test and Albert's Test. This is explained by the fact that the paper-and-pencil tests are designed to diagnose a more severe form of left-sided neglect, and many patients perform these tests with few or no errors. During paper-and-pencil testing,

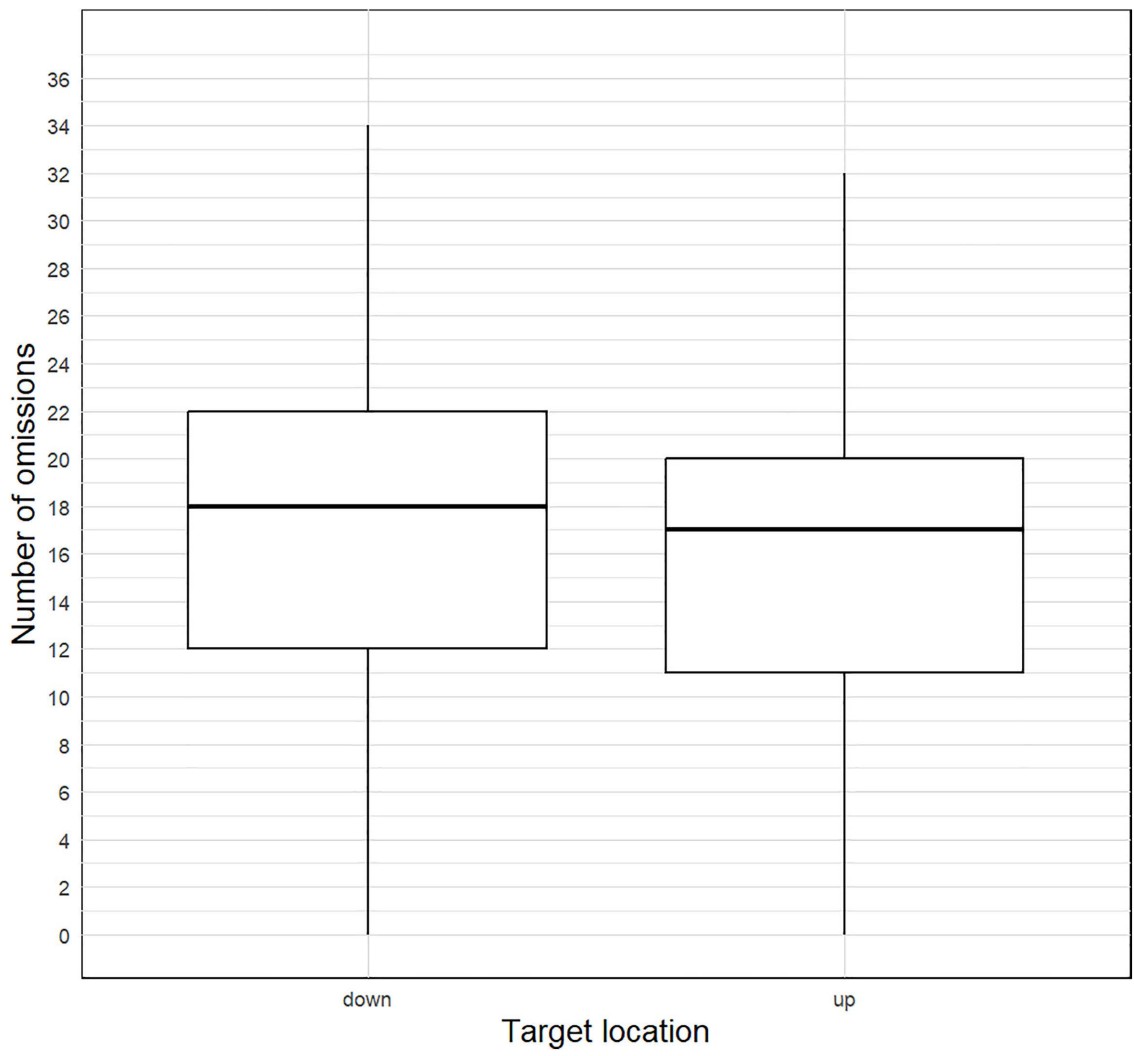

**Fig 10. Descriptive graph of vertical misses (up vs. down) in the neglect group.**

**Table 5. Sensitivity and specificity for each of the 14 target positions.**

|  | 1 | 2 | 3 | 4 | 5 | 6 | 7 | 14 | 11 | 12 | 13 | 8 | 9 | 10 |
|---|---|---|---|---|---|---|---|---|---|---|---|---|---|---|
| **Sensitivity** | 0.737 | 0.737 | 0.737 | 0.816 | 0.737 | 0.763 | 0.737 | 1 | 1 | 1 | 0.605 | 0.553 | 1 | 0.658 |
| **Specificity** | 0.609 | 0.625 | 0.609 | 0.656 | 0.594 | 0.656 | 0.484 | 0 | 0 | 0 | 0.516 | 0.484 | 0 | 0.469 |

Positions are ordered left-to-right and top-to-bottom (1–7, 14, 11–13, 8–10).

patients may use compensatory strategies, and the time to complete the tasks is not limited. In contrast, when using the computer-based method, it is virtually impossible to use compensatory strategies for visual search due to the time constraints imposed on stimulus presentation.

The primary goal in developing the computer-based Keen Eye paradigm was to overcome the low sensitivity of traditional paper-and-pencil and behavioral assessments, thereby enabling the detection of subtle and subclinical forms of

**Table 6. Correlation between Keen Eye results and paper-and-pencil tests (Bells test and Albert's test) using the Phi coefficient.**

| | Diagnosis | Model 1 | Model 2 | Model 3 | Albert's Test | The Bell's Test |
|---|---|---|---|---|---|---|
| **Diagnosis** | | | | | | |
| **Model 1** | 0.802*** | | | | | |
| **Model 2** | 0.771*** | 0.871*** | | | | |
| **Model 3** | 0.817*** | 0.924 *** | 0.943 *** | | | |
| **Albert's Test** | 0.481*** | 0.487*** | 0.512*** | 0.513*** | | |
| **The Bell's Test** | 0.515*** | 0.496*** | 0.469*** | 0.473*** | 0.570*** | |

\* p<0.05, \*\* p<0.01, \*\*\* p<0.001.

visual neglect. Unlike standard methods, which demonstrated sensitivities of 0.56 (Bells Test), 0.68 (Albert's Test), and 0.74 (Catherine Bergego Scale), Keen Eye achieved a sensitivity of 1.00 with a specificity of 0.84. Thus, the method is not intended to replicate the limited diagnostic sensitivity of conventional tests but rather to extend it by improving diagnostic power.

## Cluster analysis of patients based on Keen Eye results

A k-means cluster analysis was conducted using the parameters RSU, LSU, RSB, LSB, NIR, NIL, NIB, RH, and LH. The analysis identified three distinct patient groups (Fig 11):

1. Patients with minimal errors, demonstrating near-perfect accuracy in both circle and number identification.

2. Patients with a high number of circle omissions on the left but accurate number identification.

3. Patients with a high number of circle omissions on the left, in both unilateral and bilateral presentations, along with errors in number identification.

Thus, the Keen Eye method allows for the identification not only of a lateralized attention deficit but also of a general reduction in cognitive resources for attention allocation, which, according to studies, is also characteristic of patients with neglect [63,86].

## Relationship between attention distribution deficit across two tasks and frontal dysfunction

The identification of Cluster 1 (patients with difficulties in distributing attention across two tasks) raised the question of whether these types of errors are related to impairments in executive functions. We hypothesized that omissions or incorrect identification of digits may be associated with frontal dysfunction. However, omissions in digit identification could also result from a non-lateralized attention deficit, which is theorized to be linked to neglect severity. To test this hypothesis, we conducted a correlation analysis (Table 7) between FAB battery scores [87], neglect severity indicators assessed using the Keen Eye method (KA_bilateral, KA_all), and errors in digit identification (NIR, NIL, NIB).
The correlation analysis revealed a moderate relationship between digit identification errors and frontal dysfunction, as well as a moderate relationship between digit identification errors and the severity of neglect.

Subsequently, a non-parametric Kruskal-Wallis test was conducted to determine whether Cluster 1 differed from Clusters 2 and 3 in terms of FAB scores. Significant results were obtained using the Kruskal-Wallis test ($\chi2 = 18.1$, $p < 0.001$, $\eta^2 = 0.19$). Post hoc pairwise comparisons using the DSCF test with Holm's correction revealed significant differences between Cluster 1 and Cluster 2 (W = −5.029, p = 0.03), and between Cluster 1 and Cluster 3 (W = −5.017, p = 0.02). No significant differences were found between Clusters 2 and 3. The mean number of FAB scores obtained by patients from different clusters is presented in Fig 12.

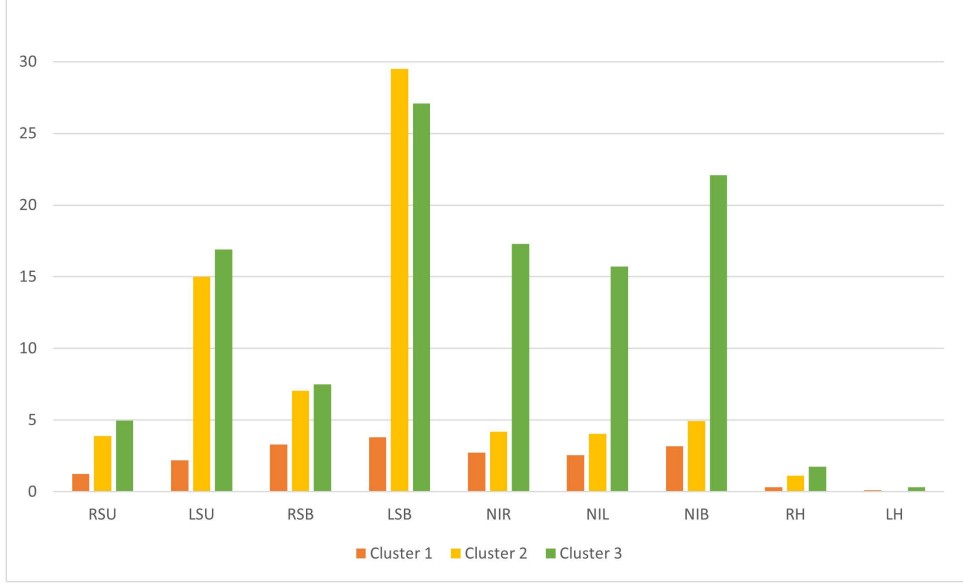

**Fig 11. Mean values of Keen Eye parameters across clusters 1, 2, and 3.**

**Table 7. Correlation between asymmetry coefficients, digit omissions, and fab scores using Spearman's Rho.**

|  | FAB | KA_bliateral | KA_all | NIR | NIL | NIB |
|---|---|---|---|---|---|---|
| **FAB** |  |  |  |  |  |  |
| **KA_bilateral** | 0.384*** |  |  |  |  |  |
| **KA_all** | 0.363*** | 0.960*** |  |  |  |  |
| **NIR** | 0.436*** | 0.401*** | 0.434*** |  |  |  |
| **NIL** | 0.339*** | 0.388*** | 0.422*** | 0.850*** |  |  |
| **NIB** | −0.327*** | 0.379*** | 0.402*** | 0.742*** | 0.690*** |  |

* p<0.05, ** p<0.01, *** p<0.001.

These pairwise comparisons demonstrate that both neglect patient clusters (Clusters 2 and 3) have significantly lower FAB scores compared to Cluster 1, but no differences were found between the two neglect clusters.

## Discussion

The Keen Eye method builds on the established dual-task framework for investigating spatial neglect, which was systematically developed by Bonato and colleagues [67–69,73,74,76–78] and is rooted in earlier concepts of dual-task interference in attention research [37,52,53]. While sharing the core principle of combining a primary spatial task with a central fixation task, the Keen Eye method extends this approach in several important ways. First, the method uses a systematically distributed set of stimulus positions whose location varied across trials but followed a predefined order, enabling detailed assessment of neglect not only along the horizontal but also the vertical axis. This design explicitly targets quadrant-specific inattention, which is underrepresented in most dual-task studies. Second, we introduce novel asymmetry coefficients to quantify the degree and direction of neglect. These coefficients provide an interpretable metric for severity and lateralization, enhancing comparability across patients. Third, unlike paradigms that primarily served

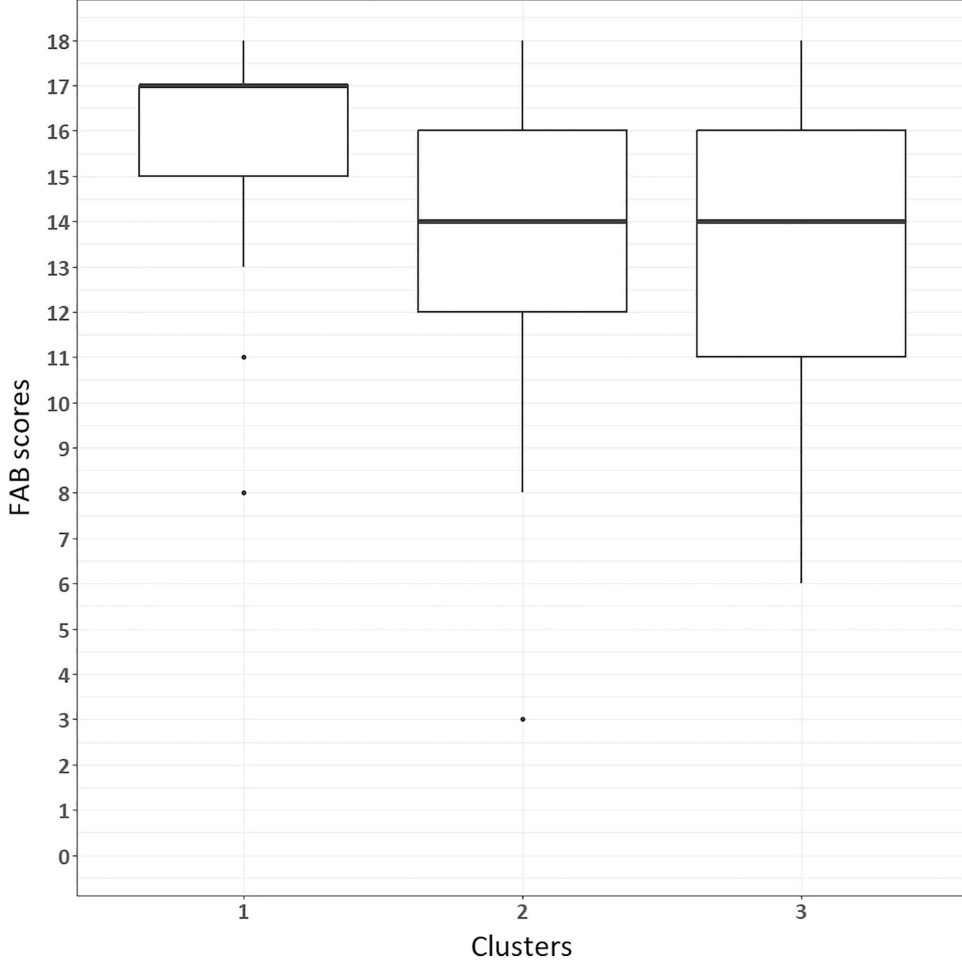

**Fig 12. FAB scores between clusters.**

experimental investigations of spatial attention, the Keen Eye method is explicitly designed for clinical diagnostic use, with standardized administration and clearly defined interpretation thresholds. Additionally, the tool was implemented entirely in open-source PsychoPy software, ensuring free accessibility and flexibility for adaptation in diverse clinical settings. Unlike some laboratory paradigms, the Keen Eye task allows for session breaks to reduce patient fatigue, and its data can be paused and resumed, if necessary, which is crucial in clinical practice. A further strength of the study is its relatively large sample size by neuropsychological standards. Because the goal of the Keen Eye method is to serve as a diagnostic tool, we deliberately recruited a broader cohort to verify the robustness and consistency of all key findings. This design choice increases confidence in the method's applicability across diverse patient populations. Taken together, these enhancements distinguish the Keen Eye method from earlier dual-task studies and address the need for a sensitive, reproducible, and clinically practical diagnostic tool.

The results obtained indicate that the Keen Eye method is suitable for detecting VN. The most diagnostically significant indicators are the omissions of circles to the left of the center of the screen during unilateral and bilateral presentations. These omissions allow for a quantitative assessment not only of inattention to single stimuli on the left but also of visual extinction phenomena. The LSB index served as our primary neglect criterion, though it primarily captures visual

extinction rather than pure neglect. In our paradigm, the maximum possible number of omissions differs between unilateral (LSU) and bilateral (LSB) conditions: specifically, LSU has a maximum of 21 trials, whereas LSB includes up to 33 bilateral presentations. This difference constrains direct comparability between these indices and must be considered when interpreting the results. We used the LSB threshold as a practical diagnostic marker because bilateral presentations are more sensitive to subtle deficits, including extinction, which often co-occurs with or masks mild neglect. By contrast, the LSU index represents more overt unilateral spatial inattention. Thus, the use of both indices provides complementary information about different stages and manifestations of spatial bias. Nonetheless, we acknowledge that future studies should refine this approach by normalizing or adjusting LSB relative to LSU, or by developing an extinction-corrected asymmetry coefficient. Such improvements would help disentangle extinction-specific effects from general neglect and enhance diagnostic precision.

Additionally, the method allows for the calculation of various asymmetry coefficients for each subject. It has been shown that the most crucial criterion for diagnosing neglect is the KA_all coefficient. This coefficient is convenient for quantitatively expressing the severity of VN, as it varies from −1–1, with a value of 0 for healthy subjects.

Thanks to the various positions of the circles relative to the horizontal line, the method can detect vertical neglect. It has been shown that patients with VN predominantly ignore the lower-left portion of the space. This finding aligns with literature data, according to which vertical neglect is often combined with horizontal neglect and primarily affects the lower part of the visual field [16,18]. This phenomenon is linked to both cortical and subcortical lesions, especially in the temporal lobe [16]. Vertical neglect can manifest as an allocentric form of spatial neglect: it is proposed that allocentric representations influence the distribution of attention resources in the vertical dimension [88]. Patients with VN demonstrate impairment in automatic, externally controlled covert attention orientation in the contralesional visual field, particularly in the lower quadrant [18]. This vertical shift, it seems, depends on time: patients require more visual fixations and more time to find a target in the lower-left quadrant [23]. This suggests that primarily involuntary automatic attention is impaired.

Our findings support the growing view that spatial neglect is not a strictly dichotomous phenomenon, but rather exists on a continuum of severity. The use of multiple quantitative indices (e.g., omission rates across different positions, asymmetry coefficients) allowed us to capture subtle attentional deficits that may not reach the clinical threshold for neglect diagnosis. This dimensional perspective is especially important for understanding borderline and subclinical cases, and for designing sensitive diagnostic tools that reflect the graded nature of visuospatial deficits.

Analysis of individual target positions revealed a pattern fully consistent with classical descriptions of neglect: patients with visual neglect predominantly omit stimuli presented on the left side of space. This leftward omission bias was stronger in the neglect group but was also detectable, to a lesser extent, in right-hemisphere patients without neglect. The robustness of contralesional omissions across both groups underlines the fundamental role of right hemisphere integrity in sustaining spatial attention. At the same time, both groups showed slightly higher omission rates for the extreme rightmost targets compared to central right targets. Previous findings indicate that patients with right-hemisphere stroke may occasionally omit targets in the ipsilesional space, especially under concurrent visual or auditory load [89]. To provide an alternative contextualization, we refer to studies on visual field organization using the irregular raster technique [85]. In these experiments, suppression of the right hemisphere resulted in perceptual frameworks centered around the central portion of the right field rather than the far right, leading to underutilization of peripheral extremes. This may explain why even patients without neglect exhibited slightly increased omissions at the rightmost positions (positions 8–10). Importantly, prior evidence also suggests that healthy individuals demonstrate higher localization accuracy in the left hemifield than in the far right, with the far right being the weakest area for spatial precision – likely reflecting right-hemisphere dominance in spatial attention [85]. Overall, our findings support a nuanced, non-linear model of neglect, while highlighting that contralesional omissions, even in patients without neglect, may serve as sensitive indicators of subtle spatial biases with direct clinical relevance.

The Keen Eye method demonstrates high sensitivity compared to existing paper-and-pencil methods that are predominantly used for neuropsychological diagnosis of VN. The Keen Eye method uncovers hidden forms of VN that are not detected during paper-and-pencil testing. Notably, it is important to emphasize that Keen Eye reduces the influence of compensatory strategies due to strict time limitations: with stimuli presented for 100 ms, patients cannot perform voluntary visual search for the target. The high sensitivity of the method is ensured by the increased load on the attention distribution system through the simultaneous presentation of two tasks. Studies show that multitasking and increased cognitive load can reveal hidden forms of spatial neglect in patients who have had a stroke. It has been shown that a dual-task paradigm, combining spatial monitoring with additional tasks, enhances the contralesional deficit in patients with damage to either hemisphere, even when standard tests show no impairments [72,73]. Such computer-based methods can uncover subclinical VN and extinction along both the horizontal and vertical axes [68,72]. The exacerbation of VN under increased cognitive load may be associated with deficits in spatial and sustained attention following damage to the right hemisphere [90].

Multitasking and high cognitive load lead to disruptions in attention processes even in healthy individuals [46,47], but these effects are more pronounced in patients with right hemisphere lesions [60,61]. It has been shown that the right hemisphere plays a crucial role in multitasking, and its damage results in a chronic deficit in distributing attention across multiple tasks, even after the regression of other symptoms [63]. Thus, limitations in multitasking are natural for a healthy brain, but in pathologies such as stroke, cognitive resources are significantly more impaired. Cognitive reserve, which typically protects against the consequences of brain damage, becomes less effective under multitasking conditions, making it possible to identify the "borderline area" between hidden and overt behavioral impairments [67]. This is especially important for diagnosing subclinical forms of VN, which can only be detected under increased cognitive load.

Moreover, the RH and LH parameters allow for the diagnosis of optical allesthesia, which is impossible in paper-and-pencil diagnostics. Optical allesthesia, or allochiria [91], is a rare phenomenon in which visual stimuli are mislocalized on the opposite side and can occur in patients with neglect. Studies have shown that increased attention load can cause false spatial localization, including allesthesia, in patients with right hemisphere damage [92]. Allochiria in VN can manifest in various modalities [93], which can be explained by a disturbance in mental spatial representation [94]. Another potential explanation is the model of attention disturbance in neglect syndrome, according to which the body-centered matrix responsible for controlling spatial attention is impaired [95]. Altered attention distribution in neglect occurs due to changes in the neural representation of egocentric space. Normally, any action in space, including voluntary control of attention via eye, head, and hand movements, is organized based on the body's position. To acquire information about the environment in an egocentric coordinate system, integration of sensory information from various spatial sources is required. The primary afferent signals related to the egocentric coordinate system are proprioceptive signals from the neck muscles and vestibular information. In patients with neglect, the process of information integration occurs with systematic errors, leading to a rotation of the entire egocentric reference system around the vertical axis to a new "equilibrium position" on the right side. It is suggested that neglect results from damage to cortical structures responsible for converting this sensory information into egocentric internal "reference matrices." It has been shown that vestibular stimulation reduces neglect symptoms [96].

Additionally, the method does not require motor responses from the subjects, allowing it to diagnose patients with motor impairments and differentiate between motor and visual forms of neglect. Research shows that with traditional paper-and-pencil tests, it is difficult to distinguish between motor and sensory neglect. It was found that neglect is typically associated with the "input channel," but the temporal and spatial aspects of neglect related to the "output channel" may require more nuanced differentiation [96]. Research also demonstrates that task-specific requirements influence the classification of attention subtypes [97]. Additionally, researchers have described differences in attention deficits between groups with perceptual and motor neglect, emphasizing the need to use both paper-and-pencil and computer-based tests to identify spatially specific and non-lateralized attention deficits [35]. Collectively, these studies highlight the complexity of assessing

neglect and the limitations of traditional tests, underscoring the need for more comprehensive and sensitive assessment methods to accurately differentiate motor and sensory neglect subtypes.

Cluster analysis of patients based on stimulus omission rates and identification errors revealed three distinct groups. One group included patients with a non-lateralized attention deficit, characterized by difficulties in distributing attention between two tasks. Another group demonstrated classic signs of VN without pronounced difficulties in attention distribution. Interestingly, patients with a non-lateralized attention deficit did not show significant differences from patients with neglect (without this deficit) on the FAB test. This may be due to the FAB evaluating a wide range of executive functions, whereas the attention distribution deficit more specifically reflects disturbances in the fronto-parietal network. Impairments in this network may not always be accompanied by significant frontal dysfunction, as measured by standard neuropsychological tests. Additionally, reduced attention to peripheral stimuli and errors in digit identification may be more closely linked to the severity of neglect rather than a separate cognitive deficit, which may also explain the absence of significant differences between the groups on the FAB.

These findings align with results from other studies, supporting the idea that damage to the right temporo-parietal region can disrupt the frontoparietal attention network, impairing the ability to maintain multiple tasks in working memory [63]. Moreover, Bayesian analysis of EEG data has revealed reduced effective connectivity between the right parietal and frontal cortex in patients with neglect, particularly when stimuli appear in the ignored field [98]. Structural neuroimaging studies further emphasize the role of white matter tracts linking the frontal and parietal cortices in spatial attention. Damage to these connections has been associated with more severe forms of neglect and reduced functional connectivity within attention networks [99–101]. In particular, lesions in the superior longitudinal fasciculus and the supramarginal gyrus have been strongly linked to chronic neglect [102]. These results suggest that non-lateralized attention deficit may be due to impaired fronto-parietal connections rather than primary frontal dysfunction. We suggest that when attention allocation deficits are detected by Keen Eye findings, it is advisable to diagnose impairment of the patient's executive functions [103]. The impairment of control functions complicates neuropsychological rehabilitation of patients with VN, as it leads to decreased criticality to their condition and difficulties in learning and using voluntary visual search strategies.

Similar effects of impaired attention distribution are also observed in conditions such as acute or chronic mental fatigue, depression, and cognitive aging. For example, studies have shown that moderate mental fatigue reduces the efficiency of attention switching between tasks, which is presumed to be related to the depletion of non-specific cognitive resources [104]. Fatigue has been shown to modulate activity in fronto-parietal and occipital regions, negatively affecting cognitive flexibility [105]. Research indicates that under high cognitive demands, both internal and external attention processes draw from a shared pool of non-specific resources [106]. Similar effects have been described in depression, where excessive rumination and divided attention overload executive control, leading to increased switch costs and impaired divided attention [107]. Likewise, aging is associated with deficits in global task switching, which correlate with reduced working memory capacity, although local switching remains relatively preserved [108]. These examples suggest that the difficulties in attention distribution observed in patients with neglect may result from a broader deficit in cognitive control mechanisms, linked to cognitive resource depletion and extending beyond spatial processing. Understanding these commonalities may provide further insight into the cognitive underpinnings of neglect and aid in developing rehabilitation strategies.

Despite its high diagnostic value, the Keen Eye methodology has some limitations. First, diagnostic accuracy may depend on equipment characteristics, such as screen size and resolution. Furthermore, task performance requires high attention concentration, which may make it difficult to use the methodology with elderly patients. Regarding clinical application, data analysis is conducted using specialized software, requiring staff training. There are also specific visual requirements for patients, as, in clinical practice, it has been difficult for patients to complete the program without glasses. Moreover, the program is unsuitable for patients with severe neurodynamic impairments, as they are unable to complete all 75 trials due to fatigue. Importantly, although Keen Eye demonstrates high classification accuracy, it should currently be regarded as a potential diagnostic instrument rather than a fully standardized clinical tool. Ongoing work is aimed at

establishing normative scoring based on healthy participants and diverse clinical populations, which will provide formal standardization and allow the method to be positioned as a validated diagnostic instrument.

Another limitation of the present study is the absence of a single-task condition, which would have allowed a direct comparison between baseline and dual-task performance. Although dual-task paradigms are more sensitive for detecting subtle forms of spatial neglect, the lack of a single-task baseline limits conclusions about the specific cognitive costs introduced by increased attentional load. Other limitation of the current version of the method is that bilateral targets are presented only in spatially symmetrical configurations. Although this design provides experimental control and reduces task complexity, it may limit ecological validity and sensitivity to subtle forms of extinction or residual neglect. Introducing asymmetrical bilateral target arrangements in future versions could help assess spatial competition in more unpredictable and naturalistic contexts, potentially revealing more nuanced attentional biases. However, adding such conditions will require careful balancing of diagnostic value and patient fatigue. Finally, a key limitation of this study is that responses to the secondary task (central digit identification) were not analyzed in detail, despite being an important criterion for clustering patients with neglect. Specifically, errors included not only incorrect digit naming but also vague responses (e.g., "I'm not sure") or reports of the digit being unseen. However, these assumptions could be tested in future work: raw response data for all 75 trials are available and may enable more in-depth analyses in follow-up studies.

Implementing this methodology into clinical practice could significantly improve neglect diagnosis. Automated data processing can reduce diagnostic time and increase result objectivity. Keen Eye could also be integrated into neurorehabilitation systems to assess patient recovery dynamics after stroke. Furthermore, the methodology could be used for the early detection of cognitive impairments and the monitoring of rehabilitation program efficacy.

For future development of the methodology, several steps are required:

1. Expanding the sample size to include testing with healthy subjects and patients with left hemisphere lesions. This would allow for refinement of diagnostic criteria and increase the specificity of the method.

2. Adapting the methodology for diagnosing neglect in other sensory modalities, such as auditory neglect.

3. Developing mobile versions of the test for convenient use in clinical settings.

The current discussion does not address how specific the methodology is for neglect and how it differentiates neglect from other perception and attention disorders. Future research should include control groups with various visual perception and attention disorders due to brain damage.

## Conclusion

The computerized Keen Eye method has demonstrated high effectiveness in diagnosing neglect, surpassing traditional tests in sensitivity. The study confirmed that the method can detect subtle manifestations of attention impairments and diagnose non-lateralized attention deficits associated with neglect. The dual-task paradigm allows for a comprehensive assessment of attention distribution. By analyzing different indicators, the method can determine the severity of neglect in both horizontal and vertical dimensions. Despite some methodological limitations, Keen Eye has significant potential for clinical application and research, offering an objective, efficient, and adaptable approach for assessing VN and related cognitive disorders.

## Supporting information

**S1 File. Database.** This table contains the raw data collected from all study participants using the Keen Eye method and related diagnostic assessments. It includes individual performance metrics such as stimulus omissions, identification errors, and asymmetry coefficients across different conditions and tasks. These data form the basis for the statistical analyses and decision tree models described in the manuscript. Each row corresponds to a single participant, with variables

representing key diagnostic indicators. On sheet 2 (Variables) in the file, explanations are provided for each variable included in the table.
(XLSX)

**S2 File. Standardized neuropsychological assessment protocol for neglect and cognitive functions.** This supplementary file presents the standardized neuropsychological assessment protocol administered to all participants in the study. The comprehensive test battery was designed to evaluate both neglect-specific deficits and general cognitive functions. For neglect assessment, the protocol included qualitative tests (copying of 5 geometric figures, table drawing, and clock drawing test) alongside quantitative measures (Albert's Test for visual neglect, Bell's Test for spatial attention, and Catherine Bergego Scale (CBS) for functional neglect evaluation). Additionally, the protocol incorporated broader cognitive assessments: the Frontal Assessment Battery (FAB) for executive functions, verbal memory tests (memorization of 6 words and 2 sentences), and a serial subtraction task (100−7) for attention and working memory. This multi-domain approach allowed for thorough characterization of neglect patterns while controlling for potential confounding cognitive deficits. The standardized administration ensured consistency across all participants, facilitating reliable comparison of results.
(PDF)

## Author contributions

**Conceptualization:** Elizaveta Vasyura.

**Data curation:** Elizaveta Vasyura.

**Formal analysis:** Elizaveta Vasyura.

**Investigation:** Elizaveta Vasyura, Georgiy Stepanov, Olga Russkikh, Daria Terentiy, Victoria Propustina.

**Methodology:** Elizaveta Vasyura, Maria Kovyazina.

**Project administration:** Yuri Zinchenko.

**Resources:** Svetlana Vasilyeva, Vadim Daminov, Yuri Zinchenko.

**Supervision:** Maria Kovyazina, Anatoliy Skvortsov, Nataliya Varako.

**Validation:** Elizaveta Vasyura.

**Writing – original draft:** Elizaveta Vasyura.

**Writing – review & editing:** Maria Kovyazina, Georgiy Stepanov, Olga Russkikh, Daria Terentiy, Victoria Propustina, Anatoliy Skvortsov, Nataliya Varako.

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
