## [Decision Letter · Decision Letter 0]

23 May 2025

Dear Dr. Vasyura,

Thank you for submitting your manuscript to PLOS ONE. After careful consideration, we feel that it has merit but does not fully meet PLOS ONE’s publication criteria as it currently stands. Therefore, we invite you to submit a revised version of the manuscript that addresses the points raised during the review process.

The manuscript presents an interesting and potentially impactful contribution to the assessment of spatial neglect using a computer-based dual-task paradigm. The results were considered to be of both scientific and clinical interest by the reviewers.

However, following evaluation by three experts, several critical issues were identified—particularly with regard to the clarity and completeness of the sample characterization and the methodological description of the task. Specific concerns were raised regarding the detailed reporting of patient characteristics, the operationalization of diagnostic criteria, and the transparency of the task’s implementation and statistical analyses. In addition, questions emerged concerning the novelty of the method in relation to previous work, which should be adequately credited, as well as some theoretical and interpretative aspects of the findings.

Despite these concerns, both the manuscript and the results were regarded as valuable. I therefore invite you to carefully address the points raised by the reviewers and to resubmit a thoroughly revised version of your manuscript for further consideration.

We look forward to receiving your revised manuscript.

Kind regards,

Giulio Contemori, Ph.D.

Academic Editor

PLOS ONE

Reviewers' comments:

Reviewer's Responses to Questions

**Comments to the Author**

1. Is the manuscript technically sound, and do the data support the conclusions?

Reviewer #1: Yes

Reviewer #2: Yes

Reviewer #3: Yes

2. Has the statistical analysis been performed appropriately and rigorously?

Reviewer #1: Yes

Reviewer #2: Yes

Reviewer #3: Yes

3. Have the authors made all data underlying the findings in their manuscript fully available?

Reviewer #1: Yes

Reviewer #2: Yes

Reviewer #3: Yes

4. Is the manuscript presented in an intelligible fashion and written in standard English?

Reviewer #1: Yes

Reviewer #2: Yes

Reviewer #3: Yes

Reviewer #1: The authors describe the validation of a test capable to assist in the diagnosis and rehabilitation of spatial neglect. The test has been administered to a large cohort of right-hemisphere stroke patients. The method leverages on the presentation of lateralized visual stimuli, which are presented at different eccentricities and possibly the lower/upper visual field. Additionally, the paradigm includes a visual dual-task: patients are asked to report a digit presented at fixation, which is known to increase the sensitivity to lateralized biases. The test has been confirmed to be sensitive to attentional biases and capable to highlight altitudinal neglect.

Overall, the topic is not entirely new. However, it bears tremendous clinical importance.

In my reading of the paper I found the task extremely similar to that described by Bonato et al. (2010, neuropsychologia). So where does the name “keen eye method” comes from? Are the authors building on previous methods called “keen eye” (not referenced) or suggesting their own? Bonato and colleagues are otherwise duly cited, including a recent preprint (now published in Communications Biology) which appears to have inspired many passages in the introduction.

My main issue with this manuscript lies in the fact that many aspects are not duly reported. In particular, the sample of patients is not described in several core clinical features. Are the patients tested in the acute, subacute, or chronic stage? This matters a lot. It would be also useful to add information about the different etiologies (i.e., the proportion of ischemic vs hemorrhagic strokes).

Importantly, the behavioral assessment leading to the diagnosis is not properly detailed as well. Later in the text it seems that both the Albert and Bells test were administered, though it remains unclear which one served the reach the diagnosis. Luria’s seminal work is referenced when mentioning that “a preliminary comprehensive neuropsychological examination to assess higher mental functions [72], as well as standardized quantitative assessments of visual neglect”. These standardized tests are what is used to assess convergence and criterion validity, which is why it is very important that all these tests are duly referenced and described early in the text for the benefit of the readers.

Other important points, in no particular order.

“Approbation” does not strike me to be a standard term. Consider revising the title and abstract to reflect the content of the paper.

Task parameters are given in terms of centimeters from the screen. Since it appears that there may be variability in the equipment used (different screen sizes and OS are mentioned), and the distance of the patients from the screen is not reported, I would advise to use degrees of visual angle as well.

Several measures in the statistical analyses are not defined or lack details: gini index, precision, recall, F1 score, all should be briefly defined for the readers. The crossvalidation setup is not explicit: how many folds were used? Which measure was used to evaluate cv performance?

Figure captions could be more descriptive. For example, figures 4,5,6 report the decision tree from three different models, but it is not spelled out the specificity of each. It would be useful to have reported the starting variables and the models’ goals (e.g., to maximize specificity) in the captions as well.

The authors write “The Keen Eye method identified LSB ≥ 12 as the threshold for diagnosing neglect, as determined by the classification tree method.” However, Figure 5 reports LSB <14. Please clarify.

As the authors acknowledge, one limitation is that the test can be long and tiring. Can the authors identify a subset of the original positions on screen such that enough sensitivity for the diagnosis is given, but with fewer trials? For example, would it be enough to administer the most eccentric positions in both the lower and upper visual fields?

Only dual-task conditions were administered. Do the authors consider useful to administer conditions without additional demands, that is a purely spatial task? For example, the authors suggest that the paradigm could be more sensitive, and identify as neglect more patients than the conventional testing. Would these patients show difficulties without multitasking? Would the definition of the clusters change when the original variables also include single task conditions?

Reviewer #2: The manuscript presents a relevant contribution to the assessment of visual neglect using a computer-based dual-task paradigm. The rationale is well grounded, and the proposed method has the potential to improve diagnostic sensitivity in both clinical and subclinical populations.

However, there are some theoretical and methodological aspects that would benefit from clarification or further elaboration. Below are specific comments and suggestions aimed at improving the clarity, rigor, and interpretability of the study.

Lines 128-129: Consider clarifying that Lavie’s Load Theory distinguishes between perceptual load and cognitive load in attentional processing. Specifically, under high perceptual load, attentional resources are fully occupied by task-relevant stimuli, thereby reducing the processing of irrelevant distractors. In contrast, under high cognitive load, top-down control may be weakened, potentially leading to increased distractor interference.

Lines 163-165: Consider adding that computer-based tasks represent a more promising method for quantifying patients’ performance for several reasons beyond the nature of the tasks themselves. First, they allow for the brief presentation of stimuli and the recording of response latencies with millisecond precision. This makes it possible to detect not only overt failures in processing contralesional space, but also subtle delays in spatial processing. Second, computerized tests can be tailored to the individual’s performance level using adaptive procedures and adjusting parameters such as stimulus duration, contrast, or spatial location until a performance threshold is reached. This reduces the risk of ceiling effects and learning effects. Furthermore, as the authors note, computerized settings facilitate the creation of tasks that engage different levels of attentional resources, up to full resource deployment under the most demanding conditions. While such attentional load can theoretically be induced even without computers, computerized implementation ensures better control, thereby enhancing diagnostic sensitivity and reproducibility.

Consider adding, at the end of the Introduction, a concise overview of the experimental method along with a clear statement of the main hypotheses. This would enhance the logical flow between the theoretical background and the empirical section of the paper. In particular, it would be helpful to briefly justify the choice of the primary task and explain why that specific secondary task was selected to manipulate attentional load.

Lines 182-189: Please clarify how the diagnosis of visual neglect was established in the VN group. It would be important to specify which specific neuropsychological tests were used. In addition, if available, please indicate how right hemisphere damage was confirmed—was neuroimaging (e.g., MRI, CT scan) performed, and were lesion locations documented or analyzed? Furthermore, consider reporting the etiology of the brain lesions (e.g., ischemic or hemorrhagic stroke, tumor), as this information is essential to fully characterize the sample.

Lines 201-206 and lines 229-230: Consider specifying in detail how many an what types of devices were used in administering the task. It is also important to indicate whether participants were seated at a fixed viewing distance from the screen, and if so, how this distance was controlled (e.g., using a chinrest, visual markers, or standardized instructions). Given that the task involves peripheral stimulus localization, variations in screen characteristics or viewing distance could significantly alter the visual angle and perceived eccentricity of stimuli, thereby impacting the task’s diagnostic validity. Providing this information would improve methodological transparency and support replicability across different research and clinical contexts.

Lines 236-240: It would be helpful to clarify how the number of trials was determined. Was this based on previous literature, pilot testing, or a formal power analysis? Given that the number of trials affects both the sensitivity of the task and the potential for participant fatigue, providing a rationale for this choice would strengthen the methodological rigor of the study.

Lines 207-216: In my view, it is important to note that a single-task condition was not included in the study design. Including such a baseline condition would have been usefull to isolate the specific effect of increased attentional load on spatial processing performance. A direct comparison between single-task and dual-task performance would have provided stronger evidence that the observed deficits emerge specifically under multitasking demands.

Lines 236-240: Currently, bilateral targets are only presented in spatially symmetrical positions. While this ensures experimental control, it may limit the ecological validity and diagnostic sensitivity of the task. Consider discussing the possibility of including asymmetrical bilateral target configurations in future versions of the method. Such arrangements could help assess the influence of spatial competition under less predictable and more naturalistic conditions, potentially revealing more subtle forms of extinction or residual neglect.

Lines 318-320: If the control group consists of the 64 patients who did not exhibit visual neglect, please specify this explicitly in the text.

Lines 479-481: Consider citing this recent study: https://doi.org/10.1038/s41598-025-95717-0

It shows that increased attentional load affects visual processing in healthy individuals, but without inducing the kind of spatial asymmetries typically observed in patients with neglect. This supports the idea that the asymmetrical effects under high attentional load are pathological rather than task-induced, and would strengthen the theoretical framework of your discussion.

Reviewer #3: In a relatively large group study with right hemisphere damaged neurological patients authors tested a computer-based method for detecting omissions of lateralized targets. Overall the method seems to have successfully detected contralesional omissions in the group of neglect patients. My main criticisms are related to the very close resemblance with previous studies and to the ambiguity in diagnostic criteria, in the analysis and in the purpose of the study.

MAIN

1) The logic behind the method as well as its implementation are very close if not almost identical to the methods that have been developed by the research group in Padova for the past 15 years (starting from Bonato et al., Neuropsychologia 2010 and Neuropsychology 2012). The resemblance is striking with respect to all aspects of the study as it includes the presence of a visual dual-task with a central digit to be reported along with lateralized targets (ipsi, contra or bilateral) whose detection accuracy is the dependent variable. The similarity extends to the clustering logic and to the interpretation in terms of compensatory strategies. While authors fully acknowledge previous studies the fact that their approach is virtually a replication of these previous studies should in my view be made explicit in the background section as well as for all the relevant sections with high resemblance (methods, analysis, discussion). Although to a lesser extent, also the idea of providing a name for the testing somehow clashes with the idea that the same method has been described in about ten previous papers and seems to result in indirect shadowing of previous studies. The aspects of novelty seem to be related to the spatial uncertainty and to the possibility to test together vertical and horizontal but paradoxically these new features are not tested explicitly nor emphasized much.

2) As for the availability of the data, many crucial aspects are missing including the etiology of the lesions (vascular only?) time from stroke and, most important, a clear operationalized set of criteria for neglect diagnosis. There is reference to two paper and pencil tests but these data are not reported nor it is clear whether all patients underwent testing with these two tests (only). More specifically, was the diagnosis for left neglect, for right neglect or for altitudinal neglect ?

3) As for the digital task, there is no indication about whether a response to the central digit was performed. If yes the handling of wrong responses should be described. If no, the ratio should be made explicit.

4) The ultimate aim of the study seems to be missing. The text refers multiple times to “high diagnostic accuracy” but this concept does not seem to have been fully operationalized. I think it would be clearer to explicitly diagnostic criteria for the different approaches (clinical vs computer based). Moreover, the task does not include a single task condition and therefore does not allow quantifying the effect of multitasking: From this point of view the statement about the presence of general impairments seems a bit far-fetched.

5) Strictly speaking LSB is an index of extinction not of neglect, and ideally should be corrected with respect to LSU. It is acceptable to keep it the way it is but this limitation should be spelled out.

6) Considering the presence of several tests (supplementary materials) I think it would be better to explicity focus on the idea that neglect from many points of view can be conceived within a continuum of omissions.

7) There are several acronyms (about 10). Due to their presence Table 3 is very difficult if not impossible to be understood. The full dataset seems full of relevant information yet the current format of table and the graphs do not seem to allow fully appreciate this info. Finally (Again in Table 3) variables 1 to 14, denoting the different positions, could be made more easy to visualize if ordered left to right.

8) Something I really couldn’t understand is the reason why positions 1,2, and 3 did not result in more omissions than positions 8, 9 and 10. For the maximal lateralization a clear disadvantage for left targets should be present. Overall, for positions 1 to 14 no clear lateralized pattern seems to be visible (a graph would help). This clashes with the conclusions and with the outcome of the clustering. Is this confusion due to the presence of trials with two targets ? From the image it seems as if there were 25 different combinations and I therefore wonder why not using all of them or, in alternative, cluster for left vs right low vs high eccentricity.

9) Clustering: at behavioural it seems unclear which are the variables better predicting neglect. The outcome of the clustering allows to refine the diagnostic threshold to converge more or less closely with the diagnosis of the clinician. Yet I see a circularity issue: if the diagnosis is based on the outcome in paper and pencil tests the outcome which is correctly predicted is exactly what we would like to avoid due to their low sensitivity. As already mentioned, the lack of criteria for describing neglect makes any reference in terms of sensitivity to the paper and pencil tests difficult to be understood.

10) The supplementary file contains a wealth of information yet several variables are not explained.

MINOR

One paper very similar to the present one is this:

https://www.nature.com/articles/s42003-025-08074-z

**Do you want your identity to be public for this peer review?** For information about this choice, including consent withdrawal, please see our Privacy Policy

Reviewer #1: No

Reviewer #2: No

Reviewer #3: No

---

## [Author Response · Author response to Decision Letter 1]

1 Jul 2025

Editor’s Comments

1. Several critical issues were identified—particularly with regard to the clarity and completeness of the sample characterization and the methodological description of the task.

Author Response: We have added a detailed description of patient characteristics in the Participants section (Lines 234–248), including information on etiology (ischemic vs. hemorrhagic), time since onset, stage of recovery (subacute or chronic), and neuroimaging confirmation of right-hemisphere lesions. We also clarified the process of clinical diagnosis of visual neglect (Lines 249–278).

2. Specific concerns were raised regarding the detailed reporting of patient characteristics, the operationalization of diagnostic criteria, and the transparency of the task’s implementation and statistical analyses.”

Author Response: We addressed this by:

• Specifying the exact clinical and neuropsychological criteria used to establish a diagnosis of neglect (Lines 249–278).

• Adding detailed descriptions of task design and implementation, including visual angle calculations, stimulus positioning, trial count rationale, and device specifications (Lines 288–362).

• Including a subsection on the decision tree modeling approach and cross-validation settings (Lines 484–513), and defining all relevant metrics (gini index, precision, recall, F1-score).

3. Questions emerged concerning the novelty of the method in relation to previous work, which should be adequately credited, as well as some theoretical and interpretative aspects of the findings.

Author Response: We clarified in the Introduction (Lines 126–134) that our dual-task approach is built on earlier theoretical and experimental traditions. In the Discussion (Lines 643–661), we explicitly differentiate the Keen Eye method from prior studies, particularly those by Bonato et al., and explain how our work extends this foundation in terms of design, clinical applicability, quadrant-level analysis, and integration of clustering-based diagnostics.

4. We note that your Data Availability Statement is currently as follows: [All relevant data are within the manuscript and its Supporting Information files.]”

Author Response: The Data Availability Statement has been updated in accordance with the requirements. We now confirm that all raw data necessary to replicate the results of the study are included in the manuscript and its Supporting Information files. Specifically:

• The complete dataset used for analyses has been provided as Supplementary File S1.

• A detailed legend describing all variable names and acronyms is included in Supplementary File S1 (Sheet 2).

• All values behind descriptive and inferential statistics, as well as those used to generate graphs and perform clustering, are fully available.

• No ethical or legal restrictions apply to data sharing in this study.

Reviewers' Comments to the Authors:

Reviewer #1

The authors describe the validation of a test capable to assist in the diagnosis and rehabilitation of spatial neglect. The test has been administered to a large cohort of right-hemisphere stroke patients. The method leverages on the presentation of lateralized visual stimuli, which are presented at different eccentricities and possibly the lower/upper visual field. Additionally, the paradigm includes a visual dual-task: patients are asked to report a digit presented at fixation, which is known to increase the sensitivity to lateralized biases. The test has been confirmed to be sensitive to attentional biases and capable to highlight altitudinal neglect.

Author response: We sincerely appreciate the reviewer's recognition of our test's clinical importance and its potential to assist in diagnosing spatial neglect.

1. In my reading of the paper I found the task extremely similar to that described by Bonato et al. (2010, neuropsychologia). So where does the name “keen eye method” comes from? Are the authors building on previous methods called “keen eye” (not referenced) or suggesting their own? Bonato and colleagues are otherwise duly cited, including a recent preprint (now published in Communications Biology) which appears to have inspired many passages in the introduction.

Author response: Thank you for your observation. We fully acknowledge the foundational role of Bonato et al.’s work in shaping the dual-task approach to diagnosing visual neglect. In response, we clarified in lines 220–222 of the Introduction that Keen Eye is our own term for a newly developed clinical diagnostic tool, inspired by—but not identical to—previous experimental designs. In lines 643–661 of the Discussion, we now explicitly outline the key innovations of the Keen Eye method in comparison to Bonato’s paradigm, including our fixed and vertically extended stimulus layout, the use of asymmetry coefficients, clinical focus, and implementation in open-source software.

2. My main issue with this manuscript lies in the fact that many aspects are not duly reported. In particular, the sample of patients is not described in several core clinical features. Are the patients tested in the acute, subacute, or chronic stage? This matters a lot. It would be also useful to add information about the different etiologies (i.e., the proportion of ischemic vs hemorrhagic strokes).

Author response: We agree this is crucial information. We have now expanded the description of the patient sample in the Participants section (lines 238–248), clearly indicating the lesion laterality, etiology (ischemic vs hemorrhagic), and stage of recovery (subacute or chronic). In addition, individual-level details for each patient are now included in Supplementary File S1.

3. Importantly, the behavioral assessment leading to the diagnosis is not properly detailed as well. Later in the text it seems that both the Albert and Bells test were administered, though it remains unclear which one served the reach the diagnosis. Luria’s seminal work is referenced when mentioning that “a preliminary comprehensive neuropsychological examination to assess higher mental functions [72], as well as standardized quantitative assessments of visual neglect”. These standardized tests are what is used to assess convergence and criterion validity, which is why it is very important that all these tests are duly referenced and described early in the text for the benefit of the readers.

Author response: Thank you for pointing this out. We revised the Participants section to clearly state how the diagnosis of visual neglect was made (lines 249–278), based on a comprehensive neuropsychological assessment. We have also added a detailed description of the full neuropsychological battery administered to all participants, now available in Supplementary File S2. This battery included both quantitative and qualitative measures and informed the reference diagnosis used for validating the Keen Eye method.

Other important points, in no particular order.

4. “Approbation” does not strike me to be a standard term. Consider revising the title and abstract to reflect the content of the paper.

Author response: We appreciate this helpful suggestion. In our native language, the term “approbation” is commonly used to refer to pilot validation of a method prior to formal standardization. However, we recognize that in English-language academic publishing, “validation” is the more appropriate and widely accepted term. Therefore, we have revised the manuscript title and abstract accordingly.

5. Task parameters are given in terms of centimeters from the screen. Since it appears that there may be variability in the equipment used (different screen sizes and OS are mentioned), and the distance of the patients from the screen is not reported, I would advise to use degrees of visual angle as well.

Author response: Thank you for this important recommendation. We have clarified the technical setup in the Materials section, including screen size, laptop model, and operating system (lines 288–294). We now report stimulus eccentricity both in centimeters and in degrees of visual angle, assuming a typical viewing distance of 60 cm (lines 343–346, 361–362). We also specify how patient distance to the screen was monitored (line 359-360) and acknowledge that formal distance fixation (e.g., chinrest) was not used due to clinical feasibility constraints.

6. Several measures in the statistical analyses are not defined or lack details: gini index, precision, recall, F1 score, all should be briefly defined for the readers. The crossvalidation setup is not explicit: how many folds were used? Which measure was used to evaluate cv performance?

Author response: Thank you for this important remark. A clear understanding of these metrics is essential for interpreting model performance. We have added concise definitions for the Gini index, precision, recall, and F1 score in lines 484–496 to help readers unfamiliar with these terms.

We also clarified the cross-validation setup: it was implemented using the rpart.control function with the xval parameter set to 10, thereby creating 10 equal-sized folds (see lines 499–513). To ensure model stability and interpretability, we specified several additional tree-growing parameters:

• minsplit = 20: minimum number of observations required for a split;

• minbucket = 7: minimum observations in terminal nodes;

• cp = 0.01: complexity parameter for pruning.

These parameters were chosen to balance model generalizability and overfitting risk, and the complexity parameter (cp) was used as the main criterion for evaluating CV performance.

7. Figure captions could be more descriptive. For example, figures 4,5,6 report the decision tree from three different models, but it is not spelled out the specificity of each. It would be useful to have reported the starting variables and the models’ goals (e.g., to maximize specificity) in the captions as well.

Author response: We fully agree that enhanced figure captions would improve clarity. Therefore, we have revised the captions for Figures 7, 8, and 9 (corresponding to Figures 4, 5, and 6 in the previous version of the manuscript) to include the model objective, list of input variables, and performance focus (e.g., balanced specificity or sensitivity or maximized sensitivity). These changes make it easier for readers to understand the logic and goals of each classification tree.

8. The authors write “The Keen Eye method identified LSB ≥ 12 as the threshold for diagnosing neglect, as determined by the classification tree method.” However, Figure 5 reports LSB <14. Please clarify.

Author response: Thank you for pointing this out. You are absolutely correct—this was a typographical error. The correct threshold determined by the classification tree is LSB <14, not ≥12. We have corrected this in the revised manuscript, specifically in line 585.

9. As the authors acknowledge, one limitation is that the test can be long and tiring. Can the authors identify a subset of the original positions on screen such that enough sensitivity for the diagnosis is given, but with fewer trials? For example, would it be enough to administer the most eccentric positions in both the lower and upper visual fields?

Author response: Thank you for raising this practical and clinically important question. We fully agree that test duration impacts patient performance and usability in real-world settings. In response, we have added a subsection titled "Diagnostic efficiency of individual target positions" to the Results section (lines 559–581), in which we evaluated the sensitivity and specificity of each stimulus location. While we are not yet ready to formally exclude any positions from the task, we discuss in this section several target locations that may be considered for future task reduction based on their diagnostic utility. These findings lay the groundwork for a more concise version of the test.

10. Only dual-task conditions were administered. Do the authors consider useful to administer conditions without additional demands, that is a purely spatial task? For example, the authors suggest that the paradigm could be more sensitive, and identify as neglect more patients than the conventional testing. Would these patients show difficulties without multitasking? Would the definition of the clusters change when the original variables also include single task conditions?

Author response: This is an excellent methodological point, and we appreciate the reviewer’s attention to the paradigm structure. We intentionally did not include a single-task condition because the primary aim of this project was to design a clinically usable diagnostic tool, prioritizing brevity, clarity, and standardization (see lines 308–316). We acknowledge, however, that the absence of a baseline single task condition limits interpretability regarding the specific effect of attentional load. We have now explicitly stated this in the Limitations section (lines 831–836). Regarding the potential impact on cluster composition: we hypothesize that including single-task performance may reduce the distinction between clusters 2 and 3 (both representing neglect), as these groups already do not differ significantly on general executive functioning (as assessed by the FAB). However, this remains speculative and would need to be empirically tested in future work.

Reviewer #2

The manuscript presents a relevant contribution to the assessment of visual neglect using a computer-based dual-task paradigm. The rationale is well grounded, and the proposed method has the potential to improve diagnostic sensitivity in both clinical and subclinical populations.

However, there are some theoretical and methodological aspects that would benefit from clarification or further elaboration. Below are specific comments and suggestions aimed at improving the clarity, rigor, and interpretability of the study.

Author response: We thank the reviewer for this thoughtful and constructive feedback. We greatly appreciate the recognition of our work’s potential clinical relevance and have addressed all points in detail below.

1. Lines 128-129: Consider clarifying that Lavie’s Load Theory distinguishes between perceptual load and cognitive load in attentional processing. Specifically, under high perceptual load, attentional resources are fully occupied by task-relevant stimuli, thereby reducing the processing of irrelevant distractors. In contrast, under high cognitive load, top-down control may be weakened, potentially leading to increased distractor interference.

Author Response: Thank you for the suggestion. We have revised the relevant section to clarify that Lavie’s Load Theory distinguishes between perceptual and cognitive load and that these two types of load exert opposite effects on distractor processing. The clarification has been added to the Introduction, lines 148–158.

2. Lines 163-165: Consider adding that computer-based tasks represent a more promising method for quantifying patients’ performance for several reasons beyond the nature of the tasks themselves. First, they allow for the brief presentation of stimuli and the recording of response latencies with millisecond precision. This makes it possible to detect not only overt failures in processing contralesional space, but also subtle delays in spatial processing. Second, computerized tests can be tailored to the individual’s performance level using adaptive procedures and adjusting parameters such as stimulus duration, contrast, or spatial location until a performance threshold is reached. This reduces the risk of ceiling effects and learning effects. Furthermore, as the authors note, computerized settings facilitate the creation of tasks that engage different levels of attentional resources, up to full resource deployment under the most demanding conditions. While such attentional load can theoretically be induced even without computers, computerized implementation ensures better control, thereby enhancing diagnostic sensitivity and reproducibility.

Author Response: Thank you for this valuable comment. We fully agree and have incorporated this elaboration into lines 103–116 of the Introduction. We emphasize the advantages of computer-based testing, including brief and precise stimulus presentation, individual task adaptability, avoidance of ceiling effect

---

## [Decision Letter · Decision Letter 1]

19 Aug 2025

Dear Dr. Vasyura,

Thank you for submitting your manuscript to PLOS ONE. After careful consideration, we feel that it has merit but does not fully meet PLOS ONE’s publication criteria as it currently stands. Therefore, we invite you to submit a revised version of the manuscript that addresses the points raised during the review process.

Thank you very much for your thorough revision of the manuscript and for addressing most of the reviewers’ concerns. The revised version has been positively evaluated, and the overall strength of the study is considered significantly improved.

However, one of the reviewers has identified a remaining major issue, as well as several minor points, that should still be addressed before the manuscript can be considered further.

We look forward to receiving your revised manuscript.

Kind regards,

Giulio Contemori, Ph.D.

Academic Editor

PLOS ONE

Journal Requirements:

Reviewers' comments:

Reviewer's Responses to Questions

**Comments to the Author**

Reviewer #1: All comments have been addressed

Reviewer #2: All comments have been addressed

Reviewer #3: (No Response)

2. Is the manuscript technically sound, and do the data support the conclusions?

Reviewer #1: Yes

Reviewer #2: Yes

Reviewer #3: Partly

3. Has the statistical analysis been performed appropriately and rigorously?

Reviewer #1: Yes

Reviewer #2: Yes

Reviewer #3: No

4. Have the authors made all data underlying the findings in their manuscript fully available?

Reviewer #1: Yes

Reviewer #2: Yes

Reviewer #3: Yes

5. Is the manuscript presented in an intelligible fashion and written in standard English?

Reviewer #1: Yes

Reviewer #2: Yes

Reviewer #3: Yes

Reviewer #1: As far as my points are concerned, I think that the authors responded very well. I would endorse this paper for publication.

Only one minor note for the authors:

Please consider revising the title again: I don’t think it is necessary to specify “in dual-task paradigm” since this is what the Keen eye method is, but if you really want to keep it consider adding an article as in “in A dual-task paradigm” or perhaps “using a dual-task paradigm”.

Best wishes

Reviewer #2: I thank the authors for their thorough and thoughtful revisions. All my previous comments have been addressed in a satisfactory manner, and the manuscript has been significantly improved.

One additional recommendation concerns the use of the term "diagnostic". While the Keen Eye method shows promising classification accuracy based on data-driven thresholds, I would advise caution in referring to it as a "diagnostic tool" in a strict clinical sense, at least until standardization procedures (e.g., normative scoring based on healthy or diverse clinical populations) are completed. A brief clarification of this point in the Discussion would be appropriate and would strengthen the positioning of the method as a potentially diagnostic instrument, pending further validation.

In summary, I have no further mandatory requests. I support the publication of this manuscript, with the optional recommendation above.

Reviewer #3: I thank the authors for having revised the manuscript according to my comments. While I think the strength of this interesting study is significantly improved I also noticed the interpretation of the findings seems to be still ambiguous. I better detail below this only major point remaining (point 4. of previous review) which in my view needs to be addressed by both the framing of the study as well as by describing and interpreting the data better.

Framing: Authors explicitly mention the clinical use of the task and should be commended in case their choice is to provide it openly to the neuropsychological community. Yet, expanding on point 4. of the first revision it still seems rather unclear whether the final aim is to achieve the same sensitivity of standard tests (paper and pencil + CBS) or whether this computer-based approach can result in different/better sensitivity. This is crucial as someone might claim that trying to replicate the (low) diagnostic sensitivity characterizing paper and pencil tests is not an optimal objective. I suggest that the reference to this aspect should be more explicit and I also suggest explaining more in depth what the task is actually measuring. This seems mandatory also when considering the very peculiar spatial distribution of omissions (Fig. 4), which seem to be more ipsilesional than contralesional.

Data description and interpretation: From the above mentioned point of view It would be clearer if the % of ipsi vs contra lesional omissions for graphs 5 would be spelled out. I would rather tend to think that the peculiar asymmetry described which favour the contralesional hemispace is due to compensation. I might be wrong but it seems prominent in patients classified as being without VN. For reference about the spatial distribution of ipsilesional omissions at different ipsilesional eccentricities authors might want to consult the study with ipsilesional targets presented with low and high eccentricity by Bonato, Romeo et al., 2019 (Frontiers). Also the asymmetries represented in figure 6 should be in my view described as indexing a bias towards ipsi or contralesional space (now this crucial aspect is not clear, in particular when considering figure 4).

MINOR

The description of the analysis is sometimes a bit more difficult than the standard. I suggest explaining the rationale/meaning more explicitly. The comprehension seems sometimes difficult also because the amount of acronyms is still overwhelming.

I suggest avoiding colloquial terms like “skeleton” for describing the spatial distribution of attention.

For neuropsychological standards the sample is rather wide, I would emphasize this strength better.

In the abstract as well as in the text target positions are referred to as “fixed”. I suggest avoiding this term as it can be easily misunderstood. Rather, I would better emphasize that the position of appearance was variable… and eventually (but in the methods only) that the targets appeared in a fixed order. The random (fixed) position of appearance is a major strength of this study along with the vertical measure. I would highlight it better as spatial uncertainty might trigger interesting mechanisms including the one I see here resulting in better contralesional performance (was this statistically tested?) which I interpret as compensatory.

I suggest checking for typos and grammatical errors.

**Do you want your identity to be public for this peer review?** For information about this choice, including consent withdrawal, please see our Privacy Policy

Reviewer #1: No

Reviewer #2: No

Reviewer #3: No

---

## [Author Response · Author response to Decision Letter 2]

31 Aug 2025

Response to Reviewers

Dear Dr. Contemori,

We would like to express our heartfelt gratitude to you and the reviewers for your invaluable time and effort in providing such thoughtful and constructive feedback on our manuscript. Your dedication to maintaining the highest scientific standards is deeply appreciated, and the review process has undoubtedly strengthened our work.

In particular, Reviewer #3’s detailed comments regarding the spatial distribution of omissions prompted us to conduct an additional quality-control check. During this process, we discovered an error in the preprocessing of target positions (1 - 14), which affected descriptive statistics, sensitivity/specificity estimates for individual positions, and Figure 4.

Importantly, the core findings of the study - including LSU, RSU, NIL, NIR, LSB, RSB, asymmetry indices, categorical decision analysis, and convergent and criterion validity results - remain fully unchanged. After correcting the preprocessing, the data now display a spatial pattern of omissions that is theoretically expected and consistent with prior literature on visual neglect. This correction makes our conclusions clearer and more robust.

The revision substantially improves consistency with established neuropsychological theories: the previously puzzling ipsilesional omissions (highlighted by Reviewer #3 as atypical) are no longer present, while the expected predominance of contralesional omissions is now clearly observed. In our view, this directly addresses Reviewer #3’s valid concerns and clarifies the interpretation of the Keen Eye method. We value transparency and scientific integrity, and we believe that reporting these corrections is essential to ensure a reliable record. To facilitate verification, we are fully prepared to provide the raw data and preprocessing script upon request.

Additionally, we have made several enhancements to strengthen the manuscript further:

• We have added CBS scores to the S1 supplementary file, enabling direct comparison of psychometric properties (sensitivity and specificity) between the CBS and our proposed Keen Eye method

• We have refined the section heading from "Convergent and criterial validity of the method" to "Psychometric properties of the method" to better reflect the expanded content, including sensitivity statistics comparing our method to traditional paper-and-pencil assessments.

Once again, we extend our sincere appreciation for your continued consideration of our work and for the rigorous peer-review process that has helped us improve this manuscript substantially.

Editor’s Comments

Authors’ Response: We greatly appreciate the reviewers’ suggestions and evaluate each recommendation carefully to ensure it adds genuine value to our work. In this revision, we have included the article recommended by Reviewer #3 in the Discussion section.

Authors’ Response: In the previous revision, we made the following updates to the reference list:

• References 37-41 were added to provide background on the historical development of dual-task paradigms in response to Reviewers.

• Reference 59 was corrected due to an earlier citation error: Zhou L, Zhen Z, Liu J, Zhou K. Neural mechanisms underlying individual differences in attentional blink. J Vis. 2019;19(10):108. doi:10.1167/19.10.108.

• Reference 67 was updated to replace a preprint with the final published article: Blini E, D'Imperio D, Romeo Z, De Filippo De Grazia M, Passarini L, Pilosio C, Meneghello F, Bonato M, Zorzi M. Susceptibility to multitasking in stroke is associated to multiple-demand system damage and leads to lateralized visuospatial deficits. Commun Biol. 2025 May 12;8(1):734. doi: 10.1038/s42003-025-08074-z.

• References 68-70, 72 were not deleted but were relocated to another section, leading to a renumbering.

• Reference 80 (Blini et al., 2016) was removed because the current study does not address right-sided spatial neglect, and this citation was no longer relevant to our sample of exclusively right-hemisphere patients.

In this revision, the only addition to the reference list is Reference 89 (Bonato et al., 2019), included following Reviewer #3’s recommendation. No other changes were made.

Reviewers' Comments to the Authors:

Reviewer #1

As far as my points are concerned, I think that the authors responded very well. I would endorse this paper for publication.

Only one minor note for the authors: Please consider revising the title again: I don’t think it is necessary to specify “in dual-task paradigm” since this is what the Keen eye method is, but if you really want to keep it consider adding an article as in “in A dual-task paradigm” or perhaps “using a dual-task paradigm”.

Best wishes

Authors’ Response: Thank you very much for your positive evaluation and for this helpful stylistic recommendation. We have revised the title and now use “using a dual-task paradigm” as suggested. We appreciate your feedback, which helped further improve the clarity of the manuscript.

Reviewer #2

Reviewer #2: I thank the authors for their thorough and thoughtful revisions. All my previous comments have been addressed in a satisfactory manner, and the manuscript has been significantly improved.

One additional recommendation concerns the use of the term "diagnostic". While the Keen Eye method shows promising classification accuracy based on data-driven thresholds, I would advise caution in referring to it as a "diagnostic tool" in a strict clinical sense, at least until standardization procedures (e.g., normative scoring based on healthy or diverse clinical populations) are completed. A brief clarification of this point in the Discussion would be appropriate and would strengthen the positioning of the method as a potentially diagnostic instrument, pending further validation.

In summary, I have no further mandatory requests. I support the publication of this manuscript, with the optional recommendation above.

Authors’ Response: Thank you for your supportive evaluation. We fully agree with your comment regarding the cautious use of the term “diagnostic tool.” In the revised manuscript, we added a clarification in the Limitations section (lines 925-930). Although the Keen Eye method demonstrates high classification accuracy based on data-driven thresholds, these criteria require further validation through standardization procedures, including the collection of normative data from healthy participants. This step is essential to ensure that predictive models do not rely solely on data from clinical samples.

We appreciate your insightful recommendation, which has strengthened the methodological positioning of our study.

Reviewer #3

Reviewer #3:

1. I thank the authors for having revised the manuscript according to my comments. While I think the strength of this interesting study is significantly improved, I also noticed the interpretation of the findings seems to be still ambiguous. I better detail below this only major point remaining (point 4. of previous review) which in my view needs to be addressed by both the framing of the study as well as by describing and interpreting the data better.

Authors’ Response: Thank you very much for this observation. We found the spatial omission pattern shown in the descriptive statistics and Figure 4 difficult and unexpected. Your renewed attention to the unsatisfactory nature of our previous explanation prompted us to perform an additional quality-control check, during which we discovered an error in preprocessing the individual target positions (1-14). Correcting this error changed the results related to individual target positions but did not affect the other indices of the method or the categorical decision analysis. We are grateful that you drew attention to this issue, which allowed us to substantially improve the accuracy and clarity of our data interpretation.

2. Framing: Authors explicitly mention the clinical use of the task and should be commended in case their choice is to provide it openly to the neuropsychological community. Yet, expanding on point 4. of the first revision it still seems rather unclear whether the final aim is to achieve the same sensitivity of standard tests (paper and pencil + CBS) or whether this computer-based approach can result in different/better sensitivity. This is crucial as someone might claim that trying to replicate the (low) diagnostic sensitivity characterizing paper and pencil tests is not an optimal objective. I suggest that the reference to this aspect should be more explicit and I also suggest explaining more in depth what the task is actually measuring.

Authors’ Response: Thank you for this valuable comment. We explicitly clarified in the Introduction that our aim is to exceed the sensitivity of classical paper-and-pencil and CBS tests rather than simply replicate it (lines 223-227). We also provided a brief description of what the Keen Eye method is designed to measure. To substantiate this aim, we calculated the sensitivity of standard paper-and-pencil tests and CBS and directly compared them with Keen Eye performance (lines 671-677).

3. This seems mandatory also when considering the very peculiar spatial distribution of omissions (Fig. 4), which seem to be more ipsilesional than contralesional.

Authors’ Response: This issue has been corrected following the preprocessing revision. The updated results now clearly show that patients with visual neglect predominantly omit contralesional stimuli and detect ipsilesional stimuli more reliably, consistent with classical descriptions of neglect.

4. Data description and interpretation: From the above mentioned point of view It would be clearer if the % of ipsi vs contra lesional omissions for graphs 5 would be spelled out.

Authors’ Response: We agree. To more accurately represent spatial bias, we reconstructed Figures 4 and 5 using the percentage of omissions rather than raw counts. This adjustment accounts for the fact that unilateral and bilateral presentations are not equally frequent and that certain positions (2, 5, 7, 14, 12, 9) occur six times whereas others occur nine times. The new figures now display the percentage of missed targets relative to the maximum possible omissions for each variable, improving clarity and comparability.

5. I would rather tend to think that the peculiar asymmetry described which favour the contralesional hemispace is due to compensation. I might be wrong but it seems prominent in patients classified as being without VN. For reference about the spatial distribution of ipsilesional omissions at different ipsilesional eccentricities authors might want to consult the study with ipsilesional targets presented with low and high eccentricity by Bonato, Romeo et al., 2019 (Frontiers). Also the asymmetries represented in figure 6 should be in my view described as indexing a bias towards ipsi or contralesional space (now this crucial aspect is not clear, in particular when considering figure 4).

Authors’ Response: Thank you for pointing this out. To quantify group differences, we compared asymmetry coefficients between patients with neglect and those without, and we additionally performed paired Wilcoxon tests comparing right- versus left-sided omissions within each group (lines 490-502). The results show a clear contralesional bias in the neglect group and only a mild bias in the control group.

We also consulted the study by Bonato et al. (2019), which confirmed that our corrected results align with previously reported spatial attention patterns.

Finally, we revised the description of Figure 6 to explicitly state that these indices reflect spatial bias toward either ipsilesional or contralesional hemispace, improving interpretability.

MINOR

6. The description of the analysis is sometimes a bit more difficult than the standard. I suggest explaining the rationale/meaning more explicitly. The comprehension seems sometimes difficult also because the amount of acronyms is still overwhelming.

Authors’ Response: We fully understand your concern regarding the large number of variables and acronyms. This comprehensive set of measures was intentionally designed to enable the most detailed analysis of the methodology’s performance. In the future, as part of standardization, it may be appropriate to limit the number of variables to those with the highest prognostic value. To improve clarity in the current manuscript, we added explicit explanations for the rationale and interpretation of key variables (lines 400-408, 420-422, 427-428, 435-436, 440-443)

7. I suggest avoiding colloquial terms like “skeleton” for describing the spatial distribution of attention.

Authors’ Response: Thank you for this note. We replaced the term “skeleton” with the more precise phrase “attentional structure” (line 768).

8. For neuropsychological standards the sample is rather wide, I would emphasize this strength better.

Authors’ Response: We agree. This important strength of our study is now explicitly highlighted in the Discussion (lines 736-740).

9. In the abstract as well as in the text target positions are referred to as “fixed”. I suggest avoiding this term as it can be easily misunderstood. Rather, I would better emphasize that the position of appearance was variable… and eventually (but in the methods only) that the targets appeared in a fixed order. The random (fixed) position of appearance is a major strength of this study along with the vertical measure. I would highlight it better as spatial uncertainty might trigger interesting mechanisms including the one I see here resulting in better contralesional performance (was this statistically tested?) which I interpret as compensatory.

Authors’ Response: Thank you for pointing this out. We revised the abstract and main text to clarify that target positions varied across trials, explicitly noting in the Methods section that they were presented in a predetermined sequence (lines 692-694, Figure 2 caption). Regarding whether better contralesional performance was statistically tested: after correcting the preprocessing of target positions, the data now demonstrate the expected predominance of contralesional omissions, which we confirmed using a Wilcoxon test.

10. I suggest checking for typos and grammatical errors.

Authors’ Response: Thank you for this reminder. We corrected all detected typos, including those in Table 3.

Thank you once again for your thoughtful consideration. Your comments substantially improved both the clarity and theoretical consistency of our manuscript.

Sincerely,

Elizaveta V. Vasyura

Lomonosov Moscow State University

vasyuraev@my.msu.ru

---

## [Decision Letter · Decision Letter 2]

14 Sep 2025

Dear Dr. Vasyura,

Thank you for submitting your manuscript to PLOS ONE. After careful consideration, we feel that it has merit but does not fully meet PLOS ONE’s publication criteria as it currently stands. Therefore, we invite you to submit a revised version of the manuscript that addresses the points raised during the review process.

**Only some minor adjustments required.**

We look forward to receiving your revised manuscript.

Kind regards,

Giulio Contemori, Ph.D.

Academic Editor

PLOS ONE

Journal Requirements:

Reviewers' comments:

Reviewer's Responses to Questions

**Comments to the Author**

Reviewer #1: All comments have been addressed

Reviewer #2: All comments have been addressed

Reviewer #3: All comments have been addressed

2. Is the manuscript technically sound, and do the data support the conclusions?

Reviewer #1: Yes

Reviewer #2: Yes

Reviewer #3: Yes

3. Has the statistical analysis been performed appropriately and rigorously?

Reviewer #1: Yes

Reviewer #2: Yes

Reviewer #3: Yes

4. Have the authors made all data underlying the findings in their manuscript fully available?

Reviewer #1: Yes

Reviewer #2: Yes

Reviewer #3: Yes

5. Is the manuscript presented in an intelligible fashion and written in standard English?

Reviewer #1: Yes

Reviewer #2: Yes

Reviewer #3: Yes

Reviewer #1: I have assessed the differences introduced in the last review round to the best of my possibilities and think that the results are sound. I commend the authors for self-correcting their work.

Reviewer #2: All my previous comments have been addressed in a satisfactory manner and I would endorse this paper for publication.

Reviewer #3: I am happy to read that my comments on the atypical spatial distribution of omissions eventually led to discover and correct an error in the data coding. Now results in general and more specifically graph 4 makes much more sense and also the rationale seems more convincing.

Some minor and final comments starting from graphs:

I could not find figure captions and therefore the values represented by the histogram seem a bit obscure. (Figure 4) Is the black line a median/average value ? Is so, does the histogram represent (Yaxis) “percentage of maximum omissions” i.e. the worst individual performance for that location ? If so please be explicit as it seems a non-standard statistic to report.

Ipsilesional omissions: I no longer see the need to discuss in depth this aspect unless there is a significant difference (maybe I missed it). My request to tackle this issue was related to the previous pattern. Now that data have been recoded and ipsilesional omissions are not particularly evident I would avoid insisting on this topic. More specifically, the lack of a control group of healthy participants does not allow to conclude much. If anything, I would instead discuss more in depth the contralesional omissions found in the N- group, as that is a result of great clinical importance.

I would also consider calling the current “control group” something more informative like “patients without neglect diagnosis”. Regardless on the specific collective name I would avoid referring to them as “control participants” as the readers might think they were healthy controls.

Signed

M Bonato

**Do you want your identity to be public for this peer review?** For information about this choice, including consent withdrawal, please see our Privacy Policy

Reviewer #1: No

Reviewer #2: No

Reviewer #3: **Yes: ** Mario Bonato

---

## [Author Response · Author response to Decision Letter 3]

18 Sep 2025

Dear Dr. Contemori,

We would like to sincerely thank you and the reviewers for the careful evaluation of our manuscript and for the constructive feedback provided at each stage of the review process. We greatly appreciate the opportunity to further revise our work. Below we provide a detailed response to all points raised. All changes in the manuscript are highlighted in the revised version with track changes.

Editor’s Comments

Author response: We carefully reviewed our reference list and confirmed that it contains no retracted papers. As requested, we also checked all references for accuracy and updated formatting where needed. One additional citation had already been included in the previous round per reviewer recommendation. During this comprehensive review, we identified and corrected several formatting inconsistencies to ensure compliance with PLOS Vancouver style requirements:

• Reference 14: Corrected the formatting of the book chapter citation to properly include "In:" before the editor name, appropriate punctuation, and "pp." for page ranges (Harvey M. Perspectives on visuospatial neglect).

• Reference 22: Updated the encyclopedia chapter formatting to use consistent page numbering format "pp. 3582-3583" instead of abbreviated form (Heilman KM, Lamb D, Williamson JB. Vertical Neglect).

• Reference 44: Corrected a typographical error in the publication year from "2917" to "2017" (Lebed AA, Korovkin SY. The unconscious nature of insight).

• Reference 47: Improved the preprint citation formatting by standardizing the month format to "March" and ensuring proper spacing around the DOI (Halder S, Raya DV, Sridharan D. Distinct neural bases of subcomponents of the attentional blink).

• Reference 69: Added the missing period at the end of the citation (Bonato M, Priftis K, Marenzi R, Umiltà C, Zorzi M. Increased attentional demands impair contralesional space awareness following stroke).

• Reference 83: Added the appropriate DOI for completeness (Azouvi P, Samuel C, Louis-Dreyfus A, et al. Sensitivity of clinical and behavioral tests of spatial neglect after right hemisphere stroke).

• Reference 99: Standardized the preprint citation formatting by spelling out "September" fully and correcting the citation date format (Dietz MJ, Nielsen JF, Roepstorff A, Garrido MI. Dysconnection of right parietal and frontal cortex in neglect syndrome).

All corrections ensure consistency with PLOS formatting requirements and improve the overall quality and accuracy of our reference list.

Reviewers' Comments to the Authors:

Reviewer #1

I have assessed the differences introduced in the last review round to the best of my possibilities and think that the results are sound. I commend the authors for self-correcting their work.

Author response: We sincerely thank you for your supportive evaluation and kind words.

Reviewer #2

All my previous comments have been addressed in a satisfactory manner and I would endorse this paper for publication.

Author Response: Thank you very much for your endorsement and encouraging feedback.

Reviewer #3

1. I am happy to read that my comments on the atypical spatial distribution of omissions eventually led to discover and correct an error in the data coding. Now results in general and more specifically graph 4 makes much more sense and also the rationale seems more convincing.

Author response: We are especially grateful for your thorough feedback, which has significantly improved the clarity and rigor of our work. Below we address your remaining minor points in detail.

Some minor and final comments starting from graphs:

2. I could not find figure captions and therefore the values represented by the histogram seem a bit obscure. (Figure 4) Is the black line a median/average value ? Is so, does the histogram represent (Yaxis) “percentage of maximum omissions” i.e. the worst individual performance for that location ? If so please be explicit as it seems a non-standard statistic to report.

Author response: Thank you for pointing this out. We have revised the captions for Figures 4–6 (lines 485-497) to explicitly clarify:

• the black horizontal line inside each box indicates the median,

• the Y-axis represents Percentage of Maximum Possible Omissions (%), calculated as the number of missed targets relative to the maximum number of presentations for each stimulus position,

• the normalization was necessary because not all positions were presented equally often (e.g., positions 2, 5, 7, 9, 12, and 14 appeared six times; all others appeared nine times),

• we also added outliers and explicitly noted that outliers are represented by dots.

3. Ipsilesional omissions: I no longer see the need to discuss in depth this aspect unless there is a significant difference (maybe I missed it). My request to tackle this issue was related to the previous pattern. Now that data have been recoded and ipsilesional omissions are not particularly evident I would avoid insisting on this topic. More specifically, the lack of a control group of healthy participants does not allow to conclude much. If anything, I would instead discuss more in depth the contralesional omissions found in the N- group, as that is a result of great clinical importance.

Author response: We agree. We have streamlined the discussion by removing extended speculation on ipsilesional omissions and shifted the focus toward the clinical relevance of contralesional omissions in patients without neglect (lines 759-778). We emphasize that even in this group, subtle left-sided deficits were detectable.

4. I would also consider calling the current “control group” something more informative like “patients without neglect diagnosis”. Regardless on the specific collective name I would avoid referring to them as “control participants” as the readers might think they were healthy controls.

Author response: Thank you for this helpful suggestion. We have replaced the term control group throughout the manuscript with patients without neglect diagnosis. We believe this improves clarity for the reader.

We are deeply grateful for the constructive guidance from the reviewers and editorial team. We believe these final adjustments have further strengthened the clarity and clinical significance of our work.

---

## [Decision Letter · Decision Letter 3]

8 Oct 2025

Validation of the Keen Eye computer-based method for diagnosing visual neglect using a dual-task paradigm

PONE-D-25-18591R3

Dear Dr. Vasyura,

We’re pleased to inform you that your manuscript has been judged scientifically suitable for publication and will be formally accepted for publication once it meets all outstanding technical requirements.

Kind regards,

Giulio Contemori, Ph.D.

Academic Editor

PLOS ONE

Additional Editor Comments (optional):

Reviewers' comments:

Reviewer's Responses to Questions

**Comments to the Author**

Reviewer #1: All comments have been addressed

Reviewer #2: All comments have been addressed

Reviewer #3: All comments have been addressed

2. Is the manuscript technically sound, and do the data support the conclusions?

Reviewer #1: Yes

Reviewer #2: Yes

Reviewer #3: (No Response)

3. Has the statistical analysis been performed appropriately and rigorously?

Reviewer #1: Yes

Reviewer #2: Yes

Reviewer #3: (No Response)

4. Have the authors made all data underlying the findings in their manuscript fully available?

Reviewer #1: Yes

Reviewer #2: Yes

Reviewer #3: (No Response)

5. Is the manuscript presented in an intelligible fashion and written in standard English?

Reviewer #1: Yes

Reviewer #2: Yes

Reviewer #3: (No Response)

Reviewer #1: I endorse this paper for publication. Best regards ______________________________________________

Reviewer #2: All my previous comments have been addressed in a satisfactory manner and I would endorse this paper for publication.

Reviewer #3: (No Response)

**Do you want your identity to be public for this peer review?** For information about this choice, including consent withdrawal, please see our Privacy Policy

Reviewer #1: No

Reviewer #2: No

Reviewer #3: **Yes: ** Mario Bonato

---

## [Editor Report · Acceptance letter]

PONE-D-25-18591R3

PLOS ONE

Dear Dr. Vasyura,

I'm pleased to inform you that your manuscript has been deemed suitable for publication in PLOS ONE. Congratulations! Your manuscript is now being handed over to our production team.

Kind regards,

on behalf of

Dr. Giulio Contemori

Academic Editor

PLOS ONE